# Integrated Experimental and Mathematical Exploration of Modular Tissue Cultures for Developmental Engineering

**DOI:** 10.3390/ijms25052987

**Published:** 2024-03-04

**Authors:** Tao Sun, Yu Xiang, Freya Turner, Xujin Bao

**Affiliations:** 1Department of Chemical Engineering, Loughborough University, Epinal Way, Loughborough LE11 3TU, UK; 2Department of Materials, Loughborough University, Epinal Way, Loughborough LE11 3TU, UK; y.xiang@lboro.ac.uk (Y.X.); x.bao@lboro.ac.uk (X.B.)

**Keywords:** developmental engineering, modular scaffold, modular tissue, human dermal fibroblast, power law model, oxygen diffusion model, 3D tissue assembly

## Abstract

Developmental engineering (DE) involves culturing various cells on modular scaffolds (MSs), yielding modular tissues (MTs) assembled into three-dimensional (3D) tissues, mimicking developmental biology. This study employs an integrated approach, merging experimental and mathematical methods to investigate the biological processes in MT cultivation and assembly. Human dermal fibroblasts (HDFs) were cultured on tissue culture plastics, poly(lactic acid) (PLA) discs with regular open structures, or spherical poly(methyl methacrylate) (PMMA) MSs, respectively. Notably, HDFs exhibited flattened spindle shapes when adhered to solid surfaces, and complex 3D structures when migrating into the structured voids of PLA discs or interstitial spaces between aggregated PMMA MSs, showcasing coordinated colonization of porous scaffolds. Empirical investigations led to power law models simulating density-dependent cell growth on solid surfaces or voids. Concurrently, a modified diffusion model was applied to simulate oxygen diffusion within tissues cultured on solid surfaces or porous structures. These mathematical models were subsequently combined to explore the influences of initial cell seeding density, culture duration, and oxygen diffusion on MT cultivation and assembly. The findings underscored the intricate interplay of factors influencing MT design for tissue assembly. The integrated approach provides insights into mechanistic aspects, informing bioprocess design for manufacturing MTs and 3D tissues in DE.

## 1. Introduction

Tissue engineering (TE) is a transformative field dedicated to the creation of biomimetic and functional tissues for medical applications. A prevailing approach in TE involves a top-down strategy, where three-dimensional (3D) in vitro tissues are reconstructed through the seeding and cultivation of mammalian cells on prefabricated 3D scaffolds. This methodology has been well established since the 1980s [1,2,3]. Despite its historical precedence, tissues manufactured using this traditional technique exhibit several inherent drawbacks, prompting the exploration of alternative strategies. One significant limitation lies in the lack of in vivo-like physiological complexity in the reconstructed tissues. This is because the conventional top-down tissue culturing protocols often imitate only the early stages of corresponding tissue regeneration processes, leading to suboptimal tissue maturity [4,5]. It also heavily relies on a limited number of dominant cell types, which are usually seeded and cultured in prefabricated 3D scaffolds. The resultant tissues lack the complex structures found in vivo and fail to replicate the physiological intricacies of mature tissues [6].

In vivo tissues exhibit highly organized structures that necessitate vascular networks and nervous systems for proper functionality. In living human tissues and organs, vascular networks play a pivotal role in supplying nutrients and removing metabolic by-products [7]. The maximal volume of approximately 0.5–1 mm^3^ is recognized as critical for efficient oxygen diffusion. This is because mass transfer limitations can lead to anoxia [8]. Confluent two-dimensional (2D) cell cultures with an oxygen consumption rate (OCR) of 3.0 × 10^−2^ mol/m^3^ s^−1^ are typically resistant to anoxia, while vascularization in 2D cell cultures are restricted by diffusion distances of no more than 5–10 μm, compared to 10–30 μm in natural mammalian tissues [8,9]. The OCR in cultured 3D tissues, ranging from 1 to 5.0 × 10^−2^ mol/m^3^ s^−1^, usually surpasses the cellular average OCR in the human body, which is 1.6 × 10^−3^ mol/m^3^ s^−1^, thereby limiting the size of reconstructed 3D tissues to approximately 100 μm [8]. However, the increase in the oxygen diffusion distance is usually unavoidable in most of the in vitro 3D tissue cultures. Tissues without vascular networks face limitations in size, with the maximum size typically restricted to <100–200 µm [10,11,12,13]. The nervous system contributes to the control of various functions, including glandular secretions and vasoconstrictions [14]. The top-down TE strategy, however, often overlooks the intricate spatial and organizational requirements of these physiological components. The resulted tissues hence fall short of replicating the complexity and functionality of their in vivo counterparts [12,13]. Additionally, top-down TE strategies encounter severe mass transfer problems during tissue cultures. These systems normally involve prefabricated solid scaffolds ranging from millimeters to centimeters and liquid medium with suspended cells, dissolved oxygen, and other nutrients. Mass transfers inside the scaffolds (e.g., cell seeding, nutrient supply, and metabolic waste removal) primarily rely on restricted diffusion [6,15,16]. In some instances, forced convections through external mechanisms, such as pumps, are employed to enhance mass transfer across the scaffolds [17,18]. However, due to the absence of vascular networks in the majority of tissues cultured using the top-down approach, the scaffold sizes are severely limited. The cell viabilities and functions particularly in the deeper regions of the scaffolds are adversely impacted [19].

These limitations have prompted a paradigm shift towards an alternative strategy known as bottom-up developmental engineering (DE). This approach aims to produce in vitro tissues with enhanced functions and physiological complexities by departing from the traditional top-down methodology [20,21,22,23]. The bottom-up DE strategy offers several distinct advantages. Foremost among these advantages is the ability to tailor physiological complexities and functions. In this novel approach, multiple cell types, including epithelial, mesenchymal, endothelial, and nerve tissue cells, are cultured on corresponding modular scaffolds (MSs) at micrometer scale. These MSs with cultured cells are then employed as modular tissues (MTs) to be assembled layer-by-layer into larger tissues. To meet the specific requirements of the temporal–spatial assembly procedures, different cell types are cultured on corresponding favorable MSs to produce MTs with varying size at different time periods. These MTs are gradually assembled, imitating the anatomic structures of the target tissues/organs. The mimic of natural development biology allows the reconstructed tissues/organs to maintain full functions with increased sizes and structure complexity. This DE strategy in fact enables the reconstruction of larger tissues with varying levels of physiological complexities [24,25]. It allows the assembly of simple tissues, such as epithelial and mesenchymal tissues, or composite tissues, such as skin and lung tissues with both epithelial and mesenchymal sections, even with the presence of vascular and/or nerve systems. Consequently, it becomes possible to replicate various stages of natural tissue development processes, offering a more nuanced and biomimetic approach to TE [26]. Moreover, this bottom-up strategy addresses the challenge of mass transfer during MT and target tissue cultures. The microscale MSs facilitate conventional diffusion for nutrient supply and metabolic waste removal. During the assembly processes, multiple MTs are gradually assembled, imitating developmental biology [6,27]. Notably, the thickness of the assembled nonvascular tissues can be controlled to <100–200 µm, ensuring sufficient mass transfer via diffusion [28]. Furthermore, the spatial assembly of MTs with various cell types includes endothelial, vascular and nerve tissue cells. The culture of these cells allows for the development of nervous and vascular networks within the target tissues. This spatial integration and organizations of various MTs during these bottom-up assembly processes enables the reproduction of natural tissue structures [29,30,31].

Despite the promising advantages of this alternative DE approach, it still remains in its infancy, and various knowledge gaps persist in MS fabrication, MT culture, target tissue assembly, and culture processes [22,32]. Critical aspects, such as the optimal materials, structures, and sizes of various MSs, initial cell seeding density on the MSs, self-aggregation of cultured MTs, suitable cell densities on individual and aggregated MTs, and optimal strategies for the assembly and culture of target tissues, are yet to be systematically investigated. The survival of cells in individual and aggregated MTs, as well as in initially assembled larger tissues without fully developed vascular networks, relies on diffusion for nutrient supply and waste removal. This reliance becomes particularly relevant during the 1–2 weeks required for vascular networks to develop for more effective nutrient perfusion [28,33,34]. Consequently, the dependence of the maximum sizes of individual and self-aggregated MTs, and the purposely assembled avascular tissues, on multiple factors (e.g., nutrient consumption rate of cultured cells, cell density, concentrations of various nutrients and metabolic waste products) needs to be thoroughly assessed. This research aims to bridge these knowledge gaps by combining experimental and mathematical methods to gain mechanistic insights into MT cultures and the assembly of target tissues. Building upon 2D cell cultures in tissue culture plastics (TCPs), tissue cultures on 3D-printed poly(lactic acid) (PLA) discs with regular open structures, and MT cultures on spherical poly(methyl methacrylate) (PMMA) MSs, power law models were developed to simulate cell growths on solid surfaces and empty spaces within 3D scaffolds. Recognizing oxygen diffusion as a primary limiting factor during 3D tissue cultures [35,36], a previously reported diffusion model [37] was adapted to simulate oxygen concentrations in cells cultured on the solid surfaces of individual spherical MSs or colonized in the empty spaces formed by aggregated MSs. These models were then combined to investigate the influences of cell density and oxygen diffusion on MT cultures and the subsequent assembly of large tissues.

Beyond providing mechanistic insights into the underlying biological processes in DE, the combination of experimental and mathematical research methods developed in this study holds considerable potential to inform the design and optimization of various bioprocesses. These processes include the large-scale culture of multiple MTs and the gradual assembly of larger functional tissues via the novel bottom-up DE strategy.

## 2. Results and Discussions

### 2.1. Investigation of Density-Dependent Cell Growths in 2D Cell Cultures

Two-dimensional cell culture on flat TCP surfaces is known for its reproducibility [38,39] and cost-effectiveness [40] compared to 3D in vitro tissue cultures. In this study, 2D cell culture was utilized to examine the density-dependent cell growths on the solid surfaces of 3D scaffolds. HDFs with varying densities (256–43,000 cells/cm^2^) were seeded in T25 flasks, reaching 90–100% confluence. Phase contrast microscopy (PCM) was used for daily cell density assessment. Figure 1IV illustrates that higher initial seeding densities (6800, 30,000, and 43,000 cells/cm^2^) led to rapid cell division, reaching confluence in 15, 12, and 8 days, respectively. Lower initial seeding density of 256 cells/cm^2^ resulted in slow cell growth until day 35, followed by accelerated cell growths, taking an additional 6 days to reach an obvious threshold density (6500 ± 500 cells/cm^2^) at day 41. Similar patterns were observed with slightly higher seeding densities of 746 and 2700 cells/cm^2^, but they reached a similar threshold in 15 and 5 days, respectively. Once the threshold density was reached, rapid cell division ensued, taking 15–16 days to achieve complete confluence (~60,000 cells/cm^2^). More repeated 2D cultures affirmed that when the initial density exceeded the threshold, HDFs exhibited rapid cell proliferation, contrasting with slow cell growths at lower initial densities. The density-dependent growth of HDFs is likely influenced by autocrine growth factors [39,41,42], extracellular matrix (ECM) components [39,42], and physical cell-to-cell contacts [39,43,44]. Fibroblasts typically produce diverse ECM components, regulating cell survival, migration, and metabolism [45,46]. Physical cell-to-cell contact has also been shown to stimulate cell proliferation [44,47,48]. Autocrine growth factors, defined as growth-promoting substances produced by cells with functional receptors, stimulate self-proliferation [49,50]. At low cell density, cell–cell contact is very limited, and minimal autocrine growth factors and ECM are produced, leading to low cell growths. These slow cell growths differ from the lag phase, characterized by initial recovery from sub-cultivations, attachment to new surfaces, and metabolic changes without cell divisions, often resulting in increased cell size rather than number [51,52]. In contrast, in the slow growth mode, cells fully attach, spread on TCP surfaces, and slowly increase in number through cell divisions. This slow growth mode is also distinct from the log phase, marked by fast binary cell fission, rapid doubling, and exponential growths [53,54]. As cell culture progresses and cell density rises, increased cell-to-cell contacts and elevated production of autocrine growth factors and ECM components prompt a shift from slow to rapid cell growths, resembling the log phase.

To establish power law cell growth models through regression analyses, the 2D cell culture data were divided into slow and rapid growth phases using the mid-value of threshold densities (6500 cells/cm^2^). Density data within the range of 6500 ± 500 cells/cm^2^ were considered as the threshold density and included in both slow and rapid growth phases. Density data equal to or greater than 60,000 cells/cm^2^ were excluded, assumed to be confluent density where cell growth halts due to cell–cell contact inhibition in 2D cell cultures [55,56]. Negative HDF growth rates were observed during the slow cell growth phases, but no cell death was evident in detailed examinations using PCM and live and dead cell staining assays. The negative growth rates could be attributed to potential deviations or errors in Image J analysis, which was used for estimating the areas covered by attached live cells and approximating cell densities in 2D cell cultures. Given the assumption of no cell death in the development of cell growth models, the exclusion of negative growth rates was verified through statistical analysis. Initially, a two-tailed F-test was applied to the 2D cell culture data, yielding F(37, 53) = 1.39, *p* = 0.05. This indicated that equal variances could be assumed in the 2D culture data with varying cell seeding densities with 95% confidence. Mean experimental data and standard deviation (SD) were calculated both with and without the negative cell growth rates. A two-tailed independent *t*-test was conducted to compare these means, revealing no significant impact of excluding negative values (t(90) = 0.477, *p* = 0.05). Consequently, the experimental data, excluding negative growth rates, were utilized to determine cell growth constants and growth orders for the slow and rapid cell growth power law models. In this study, experimental data with an initial cell seeding density of 265 cells/cm^2^ were employed to formulate the slow cell growth model: r=0.08d or $d=e^{\left(lnd_{0}+0.08t\right)}$, and the rapid cell growth model: $r={2\times 10^{9}d}^{-1.26}$ or $d=e^{(ln(4.52\times 10^{9}t+d_{0}^{2.26}))/2.26}$. These power law models were initially validated by comparing calculated and experimental data at various cell seeding densities (750, 2700, 6800, 30,000, and 43,000 cells/cm^2^), as depicted in Figure 1IV. Subsequently, these models were applied to simulate cell growths on the solid surfaces of different scaffolds.

### 2.2. Investigation of Cell Cultures on Solid Surfaces of Modular Scaffolds

The power law cell growth models were employed to predict the impacts of varying initial cell seeding densities on the culture times required for HDFs to reach confluence on the solid surfaces of PLA discs or spherical MSs. It was also used to calculate the culture times before cells started to colonize the empty spaces within these 3D scaffolds. As illustrated in Figure 2I,II, when the initial cell seeding density equaled or surpassed the threshold density and ranged from 6000 to 54,000 cells/cm^2^, cells exhibited rapid growth, and the time for achieving confluence decreased from 15 to 4 days. In contrast, with initial cell seeding densities below the threshold, ranging from 200 to 5800 cells/cm^2^, cells initially exhibited slow growth, and the time to reach the threshold density decreased from 43 to 3 days. Subsequently, the cells transitioned to rapid growth mode, requiring an additional 15–16 days to achieve confluence.

The simulation results indicate that the culture time required to produce MTs with specific target cell densities is dependent on the initial cell seeding densities. That is, the desired cell densities on MTs can be effectively adjusted by changing both the initial seeding densities and the culture time. For instance, to manufacture MTs with a cell density of 50,000 cells/cm^2^, different tissue culture processes can be devised. If the initial cell seeding density is chosen within the range of 6000–48,000 cells/cm^2^ (>threshold density), the culture time can be adjusted from 11 to 2 days accordingly. Alternatively, if the initial cell seeding density falls within the range of 200–5800 cells/cm^2^ (<threshold density), the MTs should be cultured for 43 to 2 days to reach the threshold density, followed by an additional 11 days to attain the required density of 50,000 cells/cm^2^. Consequently, the total culture time is within the range of 54 to 13 days. However, the specific parameters of MT manufacturing processes (e.g., initial cell seeding density, culture time, and final cell density) should be tailored based on the requirements of subsequent tissue assemblies.

In the bottom-up DE strategy, several critical factors need systematic exploration, including the sizes, configurations, and cell densities of multiple MSs, optimal timing for subsequent tissue assembly, thickness of the assembled nonvascular tissues, and additional culture time for vascular system development within the assembled tissues [26,57]. The survival of cells within the assembled tissues during the approximately 14-day period required for blood vessel development [7,58,59] depends on factors such as nutrient and metabolic waste diffusion rates, cell densities on the MTs, metabolic rates of cultured cells, and the thickness of assembled nonvascular tissues.

As shown in Figure 2III, when assembling MTs with a specific cell type or defined metabolic rate, utilizing MTs with cell densities below the threshold density results in cells on MT surfaces adopting a slow growth mode. As these cells gradually occupy MT surfaces, the vascular system progressively regenerates within the emerging empty spaces formed by the assembled MTs. Once blood vessels are fully developed, effective mass transfer via perfusion allows cells to become confluent not only on solid surfaces but also within the empty spaces among the MTs. Conversely, using MTs with cell densities above the threshold density (Figure 2IV) prompts cells to adapt a rapid growth mode. Quick occupation of solid surfaces and empty spaces hampers nutrient and metabolic waste diffusion due to high cell densities, adversely affecting vascular system regeneration. Insufficient mass transfer via perfusion impedes cell survival within these assembled tissues. To optimize mass transfer for cell survival, thin tissue can be assembled using MTs with high cell density or confluent cells, while thick tissue can be assembled using MTs with low cell density. To ensure cell survival in assembled nonvascular tissues, the optimal HDF density on MTs was calculated to be approximately ~30,000 cells/cm^2^ using the cell growth model. This is because, as illustrated in Figure 2I, it takes 14 days for cells to reach confluence (60,000 cells/cm^2^) on the solid surfaces of MTs. This timeframe allows the vascular system to develop adequately, supporting cells via perfusion. If MTs with HDF densities surpassing 30,000 cells/cm^2^ are used for tissue assembly, cells rapidly reach confluence on solid surfaces and colonize empty spaces between MTs before the vascular system fully develops. This adversely impacts both cell survival and vascular regeneration. Additionally, factors like cell colonization within empty spaces between the assembled MTs, metabolic rate, and the diffusion rate of various nutrients and metabolic wastes should be taken into consideration.

### 2.3. Modeling of Cell Colonization inside the Open Structures

During the slow colonization phase (3–6 days) within the regular open structures, only a few HDFs were observed. In the subsequent rapid colonization stage, a significantly larger number of cells quickly joined in the suspension bridges and coordinated to form 3D multicellular lumps with complex structures through cell stacking, as illustrated in Figure 3III(I–L). PCM and Image J analysis revealed exponential increases in cell populations within circular open pores with diameters ranging from 400 to 1100 µm (Figure 4I), corners ranging from 30 to 120 degrees (Figure 4II), and gaps ranging from 100 to 500 µm in width (Figure 4III). Interestingly, the cell populations in different regular structures did not consistently increase over the culture periods. For instance, in the open pores with a diameter of 400 µm, cell densities decreased from 5029 cells/mm^3^ at day 18 to 4607 cells/mm^3^ at day 24, then increased to 7372 cells/mm^3^ at day 30 (Figure 4I). In the corners of 30 degrees, cell densities decreased from 4611 cells/mm^3^ at day 12 to 2900 cells/mm^3^ at day 18, then increased to 4333 cells/mm^3^ at day 24, and rapidly to 18,193 cells/mm^3^ at day 30 (Figure 4II). In the gaps with a width of 100 µm, cell densities increased slowly from 133 cells/mm^3^ at day 6 to 400 cells/mm^3^ at day 12 and 433 cells/mm^3^ at day 18, then rapidly increased to 17,766 cells/mm^3^ at day 24, dropped to 14,782 cells/mm^3^ at day 30, and further increased to 22,446 cells/mm^3^ at day 36 (Figure 4III). These significant population changes could be attributed to various factors. Firstly, the processes of cell colonization within the open structures were intricate and dynamic. Continuous migrating of individual cells, and stretching and sliding of individual and multiple cellular bridges within and between the cell lumps led to constant changes in both the sizes and structures of the stacked 3D cell lumps. Secondly, some 3D cell lumps ruptured due to extreme stretching and deformation, or disturbances caused by medium changes. Thirdly, the large sizes (ranging from 100 to 500 µm) and complex structures of the 3D cell lumps, along with the limited focus depths of PCM (<100 to 200 µm), made it impossible to accurately count the cell numbers in these cell lumps. The average density within each of the precisely controlled regular structures on the PLA discs was computed using the estimated cell populations. Since the cell density during the slow colonization phase was negligible, it was not simulated. The rapid colonization phase was modeled employing Equation (7), as detailed in Section 3. In this modeling, $d_{h}$ was substituted with the cell density measured at day 6 (after the slow colonization phase), and the unit was changed from cells/cm^2^ to cells/mm^3^. As depicted in Figure 4I–III, the power law models established for the rapid cell colonization in the circular open pores, corners, and gaps were $r={10d}^{0.35}$, $r={10d}^{0.39}$, and $r={10d}^{0.46}$, respectively. To simulate cell colonization in more intricate irregular open structures, such as the empty spaces formed by aggregated spherical MSs (as illustrated in Figure 1III(D) and Figure 2III,IV), the average density within all precisely controlled regular structures on the PLA discs was computed. This density was then utilized in developing the power law model, resulting in $r={10d}^{0.39}$, as shown in Figure 4IV. This power law model was applied to approximate the time required for cells to colonize the empty spaces within the assembled tissues.

### 2.4. Investigation of Modular Tissue Cultures on Individual Spherical Modular Scaffolds

When cultivated on solid surfaces, HDFs typically form a monolayer, mirroring the behavior of many adherent mammalian cells in 2D cultures [60,61]. Our modular tissue culture experiments confirmed that HDFs cultured on the curved surfaces of individual MSs (Figure 1III(C)) displayed behaviors akin to those observed in 2D cell cultures on flat TCP surfaces, such as exhibiting spindle or fried egg shapes, random cell migration, and proliferation. For simulation purposes, HDFs cultured on spherical MSs with a radius of 100 μm were utilized, with the average thickness of fibroblasts reported at 4.7 μm [62]. The oxygen diffusion model (Equation (17)) was then integrated with the power law cell growth models to further investigate MT cultures and their impacts on subsequent MT assembly using the bottom-up DE strategy.

Assuming a 14-day period for the development of functional blood vessels within tissues [63], the same duration was considered for the vascular system’s development in tissues assembled through the DE strategy. To ensure sufficient oxygen for HDF proliferation and vascular system regeneration via oxygen diffusion, it is advisable to produce MTs with a low density of cultured HDFs for tissue assembly. This is because the oxygen concentration within the assembled tissues is inversely proportional to tissue thickness and cell densities, as illustrated in the oxygen diffusion models. As HDF density increases within the assembled tissues, the elevated oxygen requirements can be met through a combination of diffusion and perfusion via partially developed blood vessels. Upon further cultivation for 14 days, the assembled tissues achieve full confluence with HDFs, and the vascular systems become fully developed, providing essential oxygen for cell survival (Figure 3III). Consistent with predictions from the cell growth model, MTs with an HDF density of approximately 30,000 cells/cm^2^ are suitable for subsequent tissue assembly. This is because an additional 14 days is required for HDFs on MTs to reach the confluent cell density of 60,000 cells/cm^2^. During this two-week period, HDF culture within the assembled tissues relies primarily on oxygen diffusion. Once HDFs are fully confluent on MT surfaces, the fully developed vascular systems supply oxygen and other essential nutrients.

Assuming MTs with the density of 300,000 cells/cm^2^ for tissue assembly, the HDF densities on the solid surfaces of these assembled MTs were calculated after further cultivation for 7 and 14 days using the rapid cell growth model (51,621 and 60,000 cells/cm^2^). These densities were then employed to simulate oxygen levels within the HDF monolayers using the diffusion model, considering the range of HDF oxygen consumption rates from 8.3 × 10^−19^ to 1.8 × 10^−17^ moles cell^−1^ s^−1^ (see Table 1). The modeling of oxygen concentration across the HDF monolayer using Equation (17) indicated that both culture time and cell density influenced oxygen concentrations within the cell monolayers. As depicted in Figure 5I–III, utilizing the middle OCR value of the HDF and further culturing the assembled MTs for 0, 7, and 14 days resulted in a decrease in oxygen concentrations at the bottom of the cell monolayer from 199.96 to 199.93 and 199.92 µM, respectively. The OCR exhibited evident effects on oxygen concentrations within the cell monolayers. For instance, when the MTs were cultured for 7 days, using high, middle, and low values of the HDF OCR led to oxygen concentration decreases at the bottom of the cell monolayer from 199.87 to 199.93 and 199.92 µM, respectively. However, the variations in oxygen concentrations across the HDF monolayers on the MTs after 7 and 14 days of culture were nearly negligible, approximating the same oxygen concentration in the culture medium. For comparative purposes, the same densities of hepatocytes on the curved surfaces of individual spherical MSs were assumed at different culture times, and the oxygen concentrations within the hepatocyte tissues were then simulated using the OCR values ranging from 1.0 × 10^−16^ to 9.0 × 10^−16^ moles cell^−1^ s^−1^ (as shown in Table 1). Significantly higher oxygen gradients within the monolayer of hepatocytes were predicted, as shown in Figure 5IV–VI.

### 2.5. Validation of the Oxygen Diffusion Model Using Layered Tissues

To assess oxygen consumption in cell lumps within the empty spaces formed by aggregated or assembled MSs, the oxygen diffusion model (Equation (11)) designed for tissues on flat surfaces of TCPs [37] was firstly validated. This validation utilized previously published experimental data from sheets of human-endometrial-derived mesenchymal cells (hEMCs) layered to a maximum thickness of ~40 μm in Petri dishes [64]. It was detected that the minimum oxygen concentration at the base of the thickest tissue (40 μm) required for cell survival was 45% of the oxygen concentration at the top tissue surface. For validation purposes, $C_{min}=0.45\times C_{0}=90\;\mu M$ was thus assumed. The thickness of each cell sheet was measured to be 2.22 × 10^−5^ cm, with 2.5 × 10^6^ cells in each sheet, and a Petri dish diameter of 35 mm. The calculated cell density was approximately 1.169 × 10^13^ cells/L. The OCR for individual human mesenchymal stem cells (hMSCs) (see Table 1) was employed as the nearest approximation to the OCR for human hEMCs. Recognizing that some cells within tissues may be more metabolically active while others are less active or at rest, the middle OCR value for hMSCs was initially utilized for model validation. All the other constants used in the model are detailed in Table 1. The maximum tissue thickness ($T_{max}=40\;\mu m$) required to maintain the experimentally determined minimum oxygen concentration for cell survival at the base of the layered tissues [64] was firstly confirmed to be the same value as calculated using Equation (13). Subsequently, this value was validated by calculating oxygen concentrations using the middle OCR value for hMSCs in Equation (11) (Figure 6I). However, with the use of the high OCR value, the maximum tissue thickness for cell survival was reduced to approximately 20 μm. In this scenario, the oxygen concentration at the bottom of the simulated tissue (40 μm) was only 57 µM, falling below the minimal level (90 µM) required for cell survival. Conversely, when the low OCR value was employed for simulation, the oxygen concentration at the bottom of the simulated tissue increased to 125 µM, surpassing the minimal level (90 µM) necessary for cell survival. Thus, the assumption of the middle OCR value within the layered sheets of hEMCs was also validated.

### 2.6. Investigation of Tissue Assembly via the Bottom-Up Strategy

The validated oxygen diffusion model was combined with the 3D cell growth model to further examine the tissues assembled with MTs (radius: 100 μm) through the bottom-up DE strategy. For simplicity in simulations, an element volume (cube: *R* × 2 *R* × 2 *R*) within the assembled tissue was chosen. In Figure 6II,III, the radius of the MT is denoted as *R* = 100 μm, while *T* represents the thickness of the HDF tissue colonized within the empty spaces formed by two assembled MTs. The volume of the HDF tissue can be calculated using the following formula: $$V_{tissue}={4R}^{2}T-\left(\frac{1}{2}\times \frac{4}{3}{\pi R}^{3}-{\pi \left(R-T\right)}^{2}\left(R-\frac{\left(R-T\right)}{3}\right)\right).$$

Using the reported volume of fibroblasts (Table 1) and the average cell density observed in the regular structures on the PLA discs, the thickness of the tissue within the void of the volume element (VE) following the slow colonization period (3–6 days) was estimated to be approximately 15 μm. The HDF densities (cells/L) within the VE were also calculated based on the reported volume of individual cells (Table 1). It is important to note that this calculation might slightly overestimate the density, as cultured tissues typically contain intercellular gaps and extracellular matrix components [68,69]. Simulation results from the 3D cell growth model (Figure 6IV) indicated that after additional cultivation for 8 to 11 days (i.e., 14 days post-tissue assembly), the tissue thickness increased to 25 and 30 μm, respectively. Subsequently, it required another two weeks for the thickness to reach 45–50 μm and nearly 54 days to achieve 100 μm. Simulations using the oxygen diffusion model revealed that when the tissue thickness was 30 μm, the oxygen concentrations at the bottom of the tissue were 199.94, 199.39, and 198.75 μM, respectively, for low, middle, and high OCR values of fibroblasts. As the tissue thickness increased to 100 μm, the corresponding oxygen concentrations at the bottom were 199.36, 192.76, and 186.15 μM, respectively. Therefore, given the relatively low OCR values, there appears to be ample time for the development of the vascular system. Alternatively, oxygen diffusion should not adversely affect cell survival within the range of 30–100 μm thickness, meaning that HDFs colonized within the VE should receive adequate oxygen via diffusion.

To construct 3D functional tissues through the bottom-up DE strategy, it is essential to culture multiple mammalian cells on individual or aggregated MSs with varying sizes and structures [70,71]. For comparative analysis, simulations were also conducted for hepatocytes colonized within the VE using the validated oxygen diffusion model. The cell density of hepatocyte tissues, with varying thicknesses, colonized within the void of the VE was calculated based on the reported volume of hepatocytes (Table 1). Simulation results from the oxygen diffusion model revealed that the OCR values of hepatocytes had significant impacts on the oxygen concentrations at the bottom of the tissues. The use of high and middle OCR values for hepatocytes led to a significant decline in oxygen concentration at the bottom of the simulated tissues as the tissue thickness increased, as depicted in Figure 6V. Notably, due to the larger OCR values for hepatocytes compared to that for fibroblasts, hypoxic conditions could develop in these simulated tissues. Such conditions can induce the loss of cell phenotype and cause even cell death [72], especially when the oxygen concentrations reached 0 μM at tissue thicknesses of 61 and 83 μm, respectively. Consequently, cell survival at the bottom of these tissues without a developed vascular system would not be possible. When the low OCR value was used, the predicted oxygen concentration at the bottom of a 100 μm thick tissue was 141.18 μM, suggesting that cell survival could solely depend on oxygen diffusion. However, this oxygen concentration is still lower than that in simulated HDF tissues. Hence, it is crucial to consider the cell type, particularly the metabolic level, including the OCR level of the cells. For highly metabolically active cells, such as hepatocytes, MTs with low cell densities should be employed for tissue assembly. This approach could prevent hypoxic conditions within the assembled tissues. During subsequent tissue assembly, relatively thinner nonvascular tissue can be assembled to avoid hypoxic conditions before fully functional vascular systems develop.

It is noted that the oxygen diffusion model is not suitable for directly simulating oxygen concentrations within cultured 3D in vitro tissues or tissues assembled with MTs. This limitation arises because the diffusion model was designed for simulating oxygen concentrations within tissue slabs. In 3D in vitro tissues or assembled tissues, 3D scaffolds or MSs are present, which are distinct from the cultured cells. These scaffolds often possess different oxygen permeabilities, varying from impermeable to less permeable to oxygen. As a result, oxygen diffusion within the cultured 3D tissues or assembled MTs might be impeded or significantly reduced. In addition to exploring tissues assembled via the DE strategy, this study on MT cultures can potentially inform the large-scale cell cultures using suspended microcarriers [65,73]. This is because the use of spherical MSs with a diameter of 200 μm in MT cultures closely resembles the average diameter of commonly employed microcarriers for large-scale cell cultures [74,75].

## 3. Materials and Methods

### 3.1. 2D Cell Culture

Neonatal foreskin human dermal fibroblasts (HDFs, Intercytex, Manchester, UK) were cultured in TCPs such as T-flasks containing Dulbecco’s Modified Eagle’s Medium (DMEM, Lonza, Walkersville, MD, USA) supplemented with 4.5 g/L glucose, 2 mM L-glutamine (Gibco, Paisley, UK), 100 IU/mL penicillin, 100 µg/mL streptomycin (Gibco, Grand Island, NY, USA), and 10% (*v*/*v*) fetal bovine serum (FBS, Gibco, Paisley, UK). Cultures were maintained at 37 °C in a 5% CO_2_ humidified atmosphere, with media replacement every three days. Trypsin/EDTA solution (0.25% (*w*/*v*), Gibco, Paisley, UK) was used for cell detachment when reaching 80–100% confluence. After centrifugation at 300 relative centrifugal force (RCF) for 5 min, cells were resuspended in fresh medium for subsequent experiments or continuous passaging.

### 3.2. Modular Tissue Culture on Spherical Modular Scaffolds

Samples of 5 mg spherical PMMA particles with a diameter of 30–100 µm were prepared following a previously reported method [71] by dissolving PMMA in dichloromethane and injecting 8 mL of 5 mg/mL organic solution into 150 mL of 5% poly(vinyl alcohol) under magnetic stirring for 2.5 h. These particles were then added to each well of 24-well TCPs. After sterilization with 70% ethanol overnight and rinsing three times with phosphate-buffered saline (PBS, Lonza, Walkersville, MD, USA), the particles were submerged in 1 mL of DMEM medium. Some spherical MSs aggregated in DMEM, as observed in previous studies [71]. Aliquots of 1 mL of HDFs (10^3^ cells/mL) were seeded into each well and cultured at 37 °C in a 5% CO_2_ humidified atmosphere for different time periods. The media in the TCPs were replaced every three days. The cell behavior on individual and aggregated MSs during and after the MT cultures was analyzed using PCM and scanning electron microscopy (SEM), respectively.

### 3.3. Tissue Culture on PLA Discs with Finely Controlled Open Structures

PLA discs with a diameter of 6 mm and a thickness of 500 µm were fabricated using 3D printing, incorporating circular open pores (diameters: 400, 500, 640, and 1100 µm), corners (angles: 30, 60, 90, and 120°), and gaps (distances: 100, 200, 300, 400, and 500 µm), following a previously described method [71]. Briefly, the nozzle temperature of 210 °C for 3D-printing was selected, the print bed temperature was 60 °C, the printing speed was 1000 mm/min, and the set extrusion width and the layer height were 0.4 and 0.167 mm respectively. These PLA discs were sterilized with 70% ethanol overnight, washed with PBS (3 times), dried at room temperature (RT), and placed in individual wells of 24-well TCPs. Aliquots of 50 µL of HDFs (10^6^ cells/mL) were seeded onto the top surface of each PLA disc and incubated statically at 37 °C in a 5% CO_2_ humidified atmosphere for 1 h to ensure firm cell attachment. Subsequently, the discs were submerged in 1 mL of DMEM medium in TCPs and cultured for 40–60 days. The media in the 24-well TCPs were replenished every three days. The cultured cells on the PLA discs were analyzed during and after cell culture using PCM and SEM, respectively.

### 3.4. Phase Contrast and Scanning Electron Microscopy

Cells cultured on TCPs, individual and aggregated spherical MSs, or PLA discs were noninvasively monitored and analyzed using an inverted-phase contrast microscope (Eclipse TS100-F, Nikon Corporation, Shanghai, China) throughout the cultures. Cell populations were estimated by quantifying PCM images with ImageJ. After culturing, cells were gently washed with PBS (×3), fixed in intracellular fixation buffer for 10 min, thoroughly washed with distilled water, air-dried at RT, and then coated with gold/palladium (Au/Pd) for 90 s using a sputter coater (Quorum Q150R S, Laughton, UK). In situ analysis of the cells was performed using SEM (JSM-7100F FE-SEM, Singapore) in in-lens mode with a 5.0 kV accelerating beam.

### 3.5. Development of 2D Cell Growth Models

HDFs exhibited spindle shapes and increased cell density on planar TCP surfaces (Figure 1I(A–C),II(A–C)). Conversely, HDF cultures on 3D-printed PLA discs or aggregated MSs manifested two distinct phases: (i) 2D cell culture on solid surfaces, wherein HDFs adhered, migrated, and proliferated as spindle- or fried-egg-shaped cells; (ii) 3D tissue culture within open structures, as confluent 2D cultures prompted HDFs to migrate into predefined structures (e.g., corners, gaps, circular pores) on PLA discs (Figure 1I(D–F),II(D–F)) or empty spaces formed by aggregated MSs (Figure 1I(G–I),II(G–I),III(D)), inducing significant morphological changes. This observation aligns with a similar phenomenon reported in another cell culture study using microcarriers [44].

As adherent cells, HDFs established monolayers on the flat surfaces of TCPs and PLA discs, as well as the curved surfaces of spherical MSs, consistent with findings in this study and observations with other spherical particles such as microcarriers [76]. To investigate cell growths on the solid surfaces of PLA discs or spherical MSs, HDFs were cultivated on planar surfaces within T25 flasks, with daily PCM analysis of cell densities. Utilizing mass balance analysis based on 2D cell culture data, cell growth models were formulated. Given the 3-day interval for media change and the proliferative nature of HDFs in these 2D cultures, negligible cell death was assumed. Consequently, the change in cell density (*d*) per unit time (*t*) was equated to the number of newly generated cells, amenable to simulation using a power law model (Equation (1)), with *k* and *n* representing the growth constant and order of growth, respectively.
(1)$$\frac{\partial d}{\partial t}=k\cdot d^{n},$$

The cell growth rates or the differential cell densities over differential times at various time points were computed using Equation (2).
(2)$$\frac{\partial d}{\partial t}=\frac{d_{i+2}-d_{i}}{t_{i+2}-t_{i}},$$

Equation (1) was linearized (Equation (3)) for the determination of *k* and *n* by regression analysis using Excel® for Microsoft 365 MSO (Version 2311) (Microsoft, 2023).
(3)$$\ln\left(\frac{\partial d}{\partial t}\right)=a+nln(d),$$
(4)$$k=e^{a},$$

Our experiments revealed that the initial cell seeding density significantly influenced HDF growths in 2D cell culture. Figure 1IV illustrates that above a threshold density (*d*_th_) of 6500 ± 500 cells/cm^2^, HDFs exhibited rapid growth, while below *d*_th_, a slow growth mode was observed. Once the cell population surpassed *d*_th_, HDFs switched from the slow mode to rapid growth. Power law equations were developed to simulate these density-dependent slow and rapid cell growth modes.

For simulating slow cell growth, Equation (1) was integrated using *d*_l_ as the initial cell seeding density or actual cell density (<*d*_th_) on day 0, with the initiation of either cell culture or simulation, as shown in Equations (5) and (6).
(5)$$\int_{d_{l}}^{d}d^{-n}\;\partial d=\int_{0}^{t}k\;\partial t,$$
or
(6)$$d=e^{\frac{ln\left|k\;\cdot \;t\;\cdot \;\left(1-n\right)+d_{l}^{1-n}\right|}{1-n}},$$

Equation (5) is amenable for adapting to simulate rapid cell growth by substituting *d*_l_ with *d_h_* (high initial cell seeding density or actual cell density (*d_h_*)) on day 0, as demonstrated in Equation (7).
(7)$$\int_{d_{h}}^{d}d^{-n}\;\partial d=\int_{0}^{t}k\;\partial t,$$

### 3.6. Development of 3D Cell Growth Models

PCM analysis and detailed SEM examinations delineated two stages in HDF colonization within the regular structures of PLA discs or the voids created by aggregated spherical MSs: (i) Slow colonization via cellular suspension bridges, lasting 3–6 days. Upon migration into open structures, HDFs transitioned from flat spindle or fried egg shapes to 3D structures (Figure 3I). Initially, individual (Figure 3III(A–D)) or a few HDFs (Figure 3III(E–H)) continuously formed suspension bridges across small corners, gaps, or segments of regular structures. As these bridges gradually traversed from small corners toward larger empty spaces, they became highly stretched and frequently ruptured. Due to continuous bridge formation and rupture, the cell density within open pores remained relatively low and countable through PCM. Notably, this slow colonization via cellular suspension bridges exhibited similarity across different regular structures. (ii) Rapid colonization via stacked multicellular structures. With progressing cultures, more cells swiftly engaged in suspension bridges, stacking to form intricate 3D multicellular structures (Figure 3II,III(I–L)). As a consequence of this dynamic process, the complex stacked 3D cell structures not only grew in size, leading to exponential increases in cell populations, but also underwent structural alterations. These rapid colonization phases were modeled using Equation (7), with *d_h_* replaced by the cell numbers counted post-slow colonization phases; while the slow colonization phases, characterized by stable cell density, were not simulated.

### 3.7. Oxygen Diffusion Model for Tissues Cultured on Flat TCP Surfaces

Addressing the critical role of oxygen diffusion in porous scaffolds and 3D tissues during tissue cultures [35,36,77,78], the previously reported diffusion models [37] were employed to simulate oxygen gradients in cultured tissues in this study. Initially, oxygen concentrations within monolayers or stacked multilayers of cells cultured on flat surfaces in TCPs (Figure 1III(A,B)) were simulated, assuming the cultivated tissue on each flat surface was significantly wider than its thickness, making it infinitely wide. As TCP is considered impermeable to oxygen, the oxygen diffusion from the inner surfaces (including the flat bottom surface) of TCP was disregarded. Consequently, we focused solely on oxygen diffusion through the depth of the tissues, i.e., from the top surface to the tissue bottom, allowing the use of a one-dimensional model instead of a 3D model.

Utilizing Fick’s first law and mass conservation principles, Equation (8) (Fick’s second law diffusion model) was derived. This equation establishes a correlation between oxygen concentration (*C*), tissue culture time (*t*), oxygen diffusion coefficient (*D*), and the location inside the cultured tissue (*x*) [79].
(8)$$\frac{\partial C}{\partial t}=D\frac{\partial^{2}C}{\partial x^{2}},$$

In Equation (9), a metabolic component (*φ*) was introduced to account for the consumption rate of a specific nutrient, such as oxygen, by live cells [37].
(9)$$\frac{\partial C}{\partial t}+\varphi =D\frac{\partial^{2}C}{\partial x^{2}},$$

In situations where cells have ceased division or are in a stable metabolic state, *φ* can be considered constant. Equation (10) defines *φ* specifically for oxygen.
(10)φ=OCR×cells per liter,
where *OCR* is oxygen consumption rate, and cells per liter is cell density. 

Equation (11) provides a means to determine the oxygen concentration C(x) at certain depth (x) within the simulated tissue [37]. Here, *T* represents the tissue thickness. The initial oxygen concentration at the tissue surface is denoted as $C\left(x=0\right)=C_{0}$.
(11)$$C\left(x\right)=\frac{\varphi x^{2}}{2D}-\frac{\varphi Tx}{D}+C_{0},$$

This diffusion equation relies on certain assumptions. Firstly, it assumes that the simulated tissue reaches an equilibrium or quasi-steady state, implying $\frac{\partial C}{\partial t}=0$. That is, the oxygen concentrations at particular locations inside the tissue remans do not change with time at the moment of simulation. Secondly, there is no change in oxygen concentration at the tissue bottom, i.e., $\frac{\partial C}{\partial x}(x=T,t)=0$. Thirdly, the oxygen concentration at the tissue surface is a constant, $C(x=0,t)=C_{0}$, representing the dissolved oxygen concentration in the culture media. Additionally, it is presumed that the media volume is sufficiently large, ensuring constant dissolved oxygen in the media, which is not changed by the culture cells. 

Theoretically, if the concentration of essential nutrients at the deepest part of the tissue is 0, or $C\left(x\right)=0$ when x=T, the maximum tissue thickness can be calculated using Equation (12).
(12)$$T_{max}=\sqrt{\frac{2C_{0}D}{\varphi }},$$

In cases where the oxygen concentration at the deepest part of the tissue is 0, cells at the tissue bottom face the risk of starvation. Hence, the maximum tissue thickness (x=T) is defined when the oxygen concentration reaches a critical threshold $\left(C\left(x\right)=C_{min}\right)$, where $C_{min}$ represents the minimum oxygen concentration required for cell survival. A more realistic determination of the maximum tissue thickness can be achieved using Equation (13).
(13)$$T_{max}=\sqrt{\frac{2D(C_{0}-C_{min})}{\varphi }},$$

In this study, we employed Equation (11) to model the diffusion of oxygen through the HDFs cultured on the flat surfaces of TCPs or the top surfaces of PLA discs.

### 3.8. Oxygen Diffusion Model for Cells Cultured on Spherical Modular Scaffolds

To simulate oxygen diffusion through spherical cellular structures or organoids, such as cerebral organoids, Equation (14) was derived by further modifying Equation (11) [37].
(14)$$C\left(r\right)=\frac{\varphi }{6D}\left(r^{2}-R^{2}\right)+C_{0},$$
where *R* is the radius of spherical organoid, and *r* is the distance from the center to a point within the sphere. 

This model relies on several assumptions. Firstly, both cell density and oxygen diffusion coefficient are considered constant within the cellular sphere. Secondly, the size or dimension of the sphere is assumed to remain constant, with no shrinking or swelling. Thirdly, the oxygen concentration at the center of the sphere is assumed to be constant, or $\frac{\partial C(r=0)}{\partial r}=0$. Finally, the concentration at the outer surface of the sphere (i.e., the outer edge of the tissue) is given by a constant, $C(r=R,t)=C_{0}$.

In this study, Equation (14) was further modified to simulate oxygen diffusion through cells cultured on the curved surfaces of spherical MSs or microcarriers (as depicted in Figure 1III(C)). Equation (15) illustrates the modification, where *R* was changed from the organoid radius to the radius of a spherical MS or a microcarrier plus the thickness of the cultured tissue.
(15)$$R=R_{MS}+T_{cell\;layer},$$

Furthermore, the variable r was substituted with the term *x* (as shown in Equation (16)), representing the distance from the tissue surface to a point within the modeled sphere (Figure 1III(C)). This adjustment ensures consistency with other diffusion models, all of which measure distances from tissue surfaces to points within the tissues.
(16)r=R−x,

Based on the aforementioned modifications, Equation (17) was derived. It is essential to note that this model is specifically designed for simulating oxygen diffusion through cultured tissues, not within the spherical MSs or microcarriers. Therefore, its validity is confined to the range of $0\leq x\leq T_{cell\;layer}$.
(17)$$C\left(x\right)=\frac{\varphi }{6D}\left({\left(R-x\right)}^{2}-R^{2}\right)+C_{0},$$

### 3.9. Summary of The Models Used in This Research

(1)2D Cell Growth Models(i)Slow Cell Growth Model:$$\int_{d_{l}}^{d}d^{-n}\;\partial d=\int_{0}^{t}k\;\partial t$$(ii)Rapid Cell Growth Model: $$\int_{d_{h}}^{d}d^{-n}\;\partial d=\int_{0}^{t}k\;\partial t$$
where *t* is culture time, *d* is cell density, *d*_l_ is the initial low cell seeding density, *d_h_* is the initial high cell seeding density, *k* is the cell growth constant, and *n* is order of cell growth.(2)3D Cell Growth Model:$$\int_{d_{h}}^{d}d^{-n}\;\partial d=\int_{0}^{t}k\;\partial t$$
where *t* is culture time, *d* is cell density, *d_h_* is the cell numbers colonized in the open pores after the slow colonization phases, *k* is the cell growth constant, and *n* is order of cell growth.(3)Oxygen Diffusion Model for Tissues Cultured on Flat TCP Surfaces:$$C\left(x\right)=\frac{\varphi x^{2}}{2D}-\frac{\varphi Tx}{D}+C_{0}$$
where *x* is the location inside the simulated tissue, *T* is the maximum tissue thickness, *φ* is the oxygen consumption rate, *D* is the oxygen diffusion coefficient, C(x) is the oxygen concentration at the depth of (x) within the simulated tissue, and $C_{0}$ is the initial oxygen concentration at the tissue surface, i.e., as $C_{0}=C\left(x=0\right)$.(4)Oxygen Diffusion Model for Cells Cultured on Spherical Modular Scaffolds:$$C\left(x\right)=\frac{\varphi }{6D}\left({\left(R-x\right)}^{2}-R^{2}\right)+C_{0}$$
where *R* is the radius of spherical modular scaffold (or the microcarrier) plus the thickness of the cultured tissue, *x* is the location inside the simulated tissue, *φ* is the oxygen consumption rate, *D* is the oxygen diffusion coefficient, C(x) is the oxygen concentration at the depth of (x) within the simulated tissue, and $C_{0}$ is the initial oxygen concentration at the tissue surface, i.e., as $C_{0}=C\left(x=0\right)$.

### 3.10. Statistical Analysis

All the experimental results are shown as mean ± SD from at least three independent replicate experiments (*n* ≥ 3). One-way analysis of variance (ANOVA) was used for statistical significance analysis (* *p* < 0.05, ** *p* < 0.01, *** *p* < 0.001). For the development of the power law cell growth models, a two-tailed F-test was firstly used to confirm that equal variances could be assumed in the 2D culture data with varying cell seeding densities; a two-tailed independent *t*-test was then conducted to compare the mean values of the 2D culture data for each cell seeding density with or without excluding the negative cell growth rates.

## 4. Conclusions

Our experiments revealed that HDFs exhibited typical 2D monolayer cell cultures on the solid surfaces of different scaffolds. However, when HDFs migrated into the regular structures on PLA discs or the empty spaces formed between aggregated MSs, they transitioned into 3D cultures with distinct cell morphologies and coordinated behaviors. Power law models were developed based on cell and tissue culture data to simulate the dependence of cell growth on cell densities and culture time periods. We adapted a diffusion model to simulate oxygen diffusion within cells cultured on the curved surfaces of individual MSs or in the empty spaces formed by aggregated MSs. These models were combined to investigate modular tissue cultures and subsequent tissue assembly processes. The results highlight the importance of manufacturing MTs with suitable cell densities. The considered factors were cell type, metabolic rate, initial seeding density, size and structure of individual and aggregated MSs, nutrient diffusion, and culture time period. Beyond providing insights into MT culture and assembly, our integrated experimental and computational methods could potentially inform bioprocess design for large-scale MT manufacture. These methods also propose suitable strategies for assembling MTs into more functional 3D tissues in DE. Additionally, given the similarities in size and function between spherical MSs and microcarriers, the research methods developed here could be applied to bioprocess design for microcarrier-based large-scale cell expansions.

## Figures and Tables

**Figure 1 ijms-25-02987-f001:**
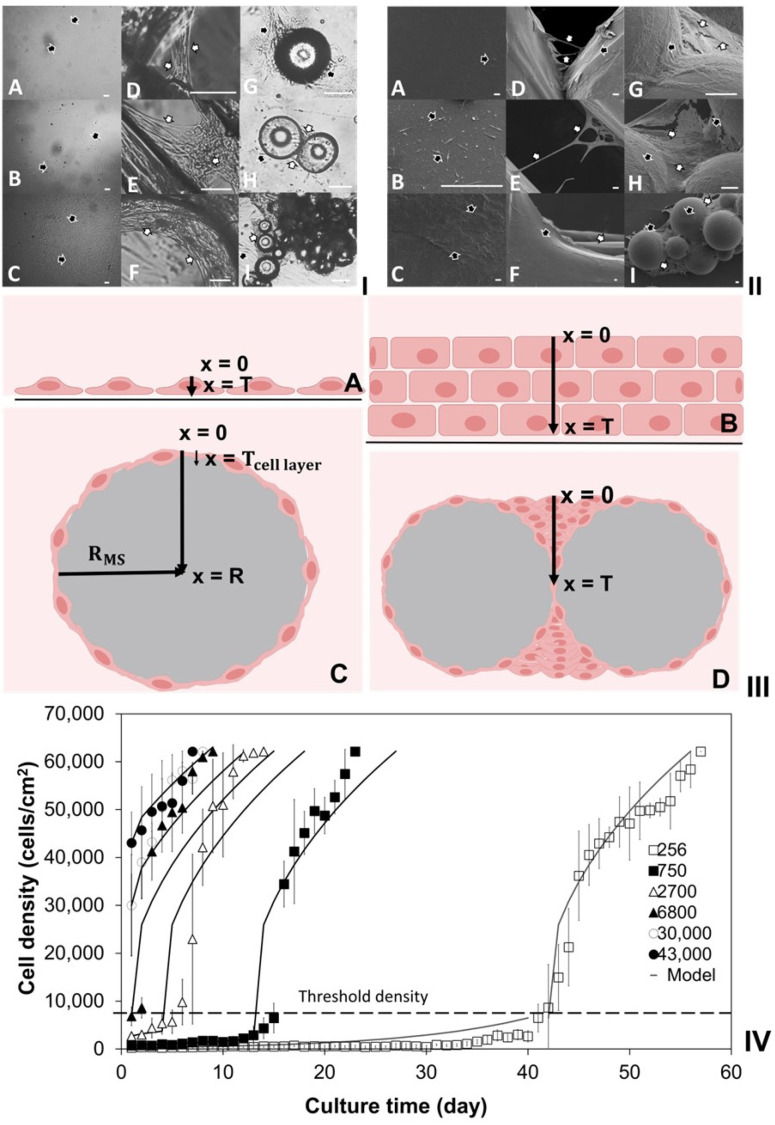
(**I**) Phase contrast micrographs (scale bars = 100 µm); (**II**) scanning electron micrographs (scale bars = 10 µm) of the human dermal fibroblasts (HDFs) cultured on (**A**–**C**) tissue culture plastics (TCPs), (**D**–**F**) finely controlled regular open structures of poly(lactic acid) (PLA) discs, (**G**–**I**) aggregated spherical modular scaffolds (MSs). HDFs on solid surfaces or open spaces are highlighted using black and white arrows, respectively. (**III**) Schematic diagrams of cells cultured on TCPs, individual and aggregated spherical MSs. Cross-sections of (**A**) cell monolayer or (**B**) stacked cells on the flat surfaces of TCPs; (**C**) cell monolayer on the curved surface of individual MS; (**D**) cells colonized in the open space between aggregated MSs, respectively, where x is the depth into the tissue ranging from x = 0 at the surface to x = T at the bottom of the tissue. (**IV**) Growths of HDFs on the flat surfaces of TCPs with varying initial cell seeding densities (□: 256, ■: 750, △: 2700, ▲: 6800, ○: 30,000, ●: 43,000 cells/cm^2^). The density-dependent slow and fast cell growths segmented by the threshold density (dotted line: 6500 ± 500 cells/cm^2^) were simulated via power law models (solid line).

**Figure 2 ijms-25-02987-f002:**
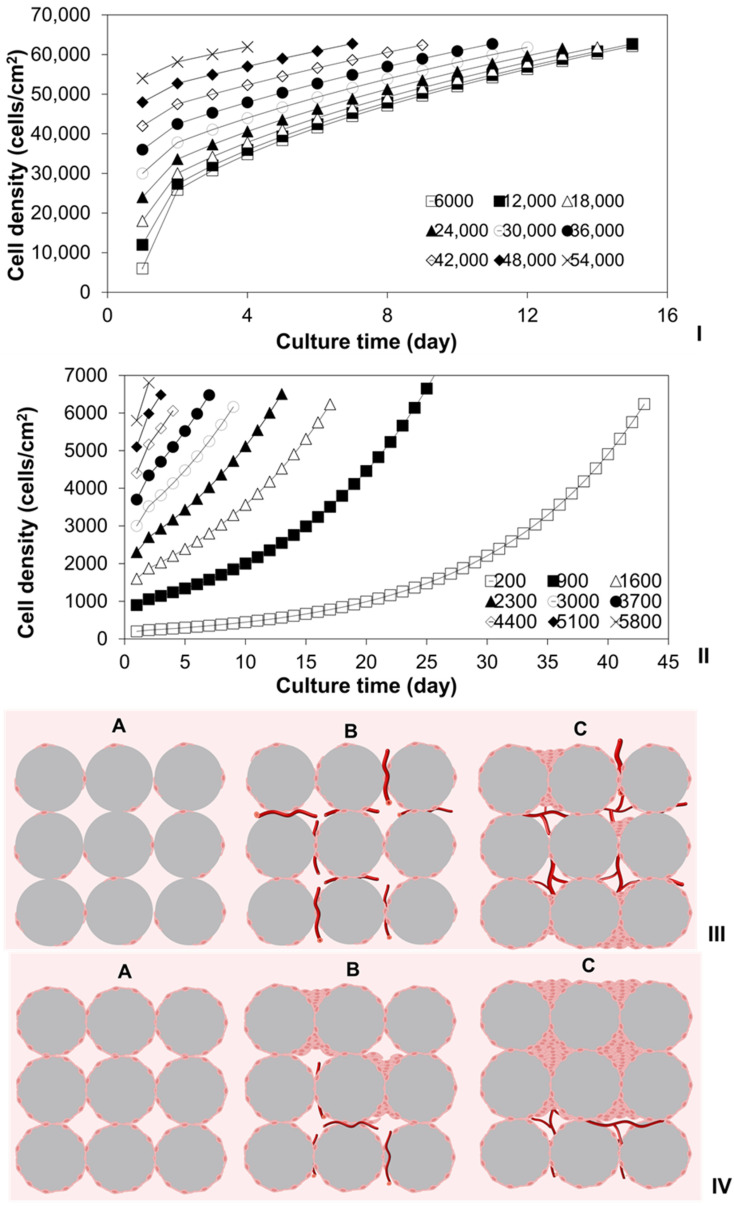
Simulation of (**I**) rapid growths of HDFs cultured on TCPs with varying initial densities (□: 6000, ■: 12,000, △: 18,000, ▲: 24,000, ○: 30,000, ●: 36,000, ◇: 42,000, ◆: 48,000, ×: 54,000 cells/cm^2^); (**II**) slow growths of HDFs cultured on TCPs with varying initial densities (□: 200, ■: 900, △: 1600, ▲: 2300, ○: 3000, ●: 3700, ◇: 4400, ◆: 5100, ×: 5800 cells/cm^2^). Schematic diagrams of vascular regenerations in tissues assembled using spherical modular tissues with (**III**) low or (**IV**) high density of HDFs ((**A**) early stage for the culture of the assembled tissues, (**B**) initialization of blood vessel formation within the assembled tissues after cultured for less than 1–2 weeks, (**C**) vascular regeneration within the assembled tissues after cultured for 1–2 weeks).

**Figure 3 ijms-25-02987-f003:**
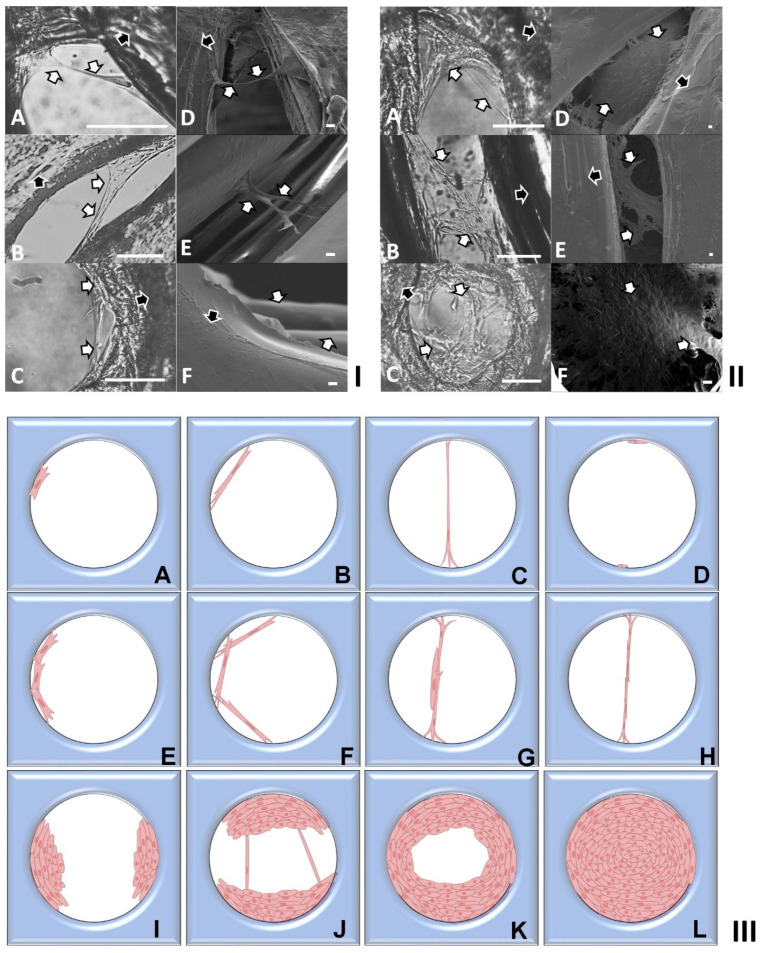
(**I**,**II**) Micrographs of HDFs colonized in the finely controlled regular open structures of PLA discs in (**I**) early slow colonization stage (3–6 days) via cell bridging; and (**II**) the fast colonization stage via cell stacking and bridging (as highlighted via white and black arrows); (**A**–**C**): phase contrast micrographs (scale bars = 100 µm); (**D**–**F**): scanning electron micrographs (scale bars = 10 µm). (**III**) Schematic diagrams of HDFs colonized in the finely controlled regular open structures of PLA discs in (**A**–**H**) early slow colonization stage (3–6 days) via cell bridging or (**I**–**L**) fast colonization stage via cell stacking and bridging.

**Figure 4 ijms-25-02987-f004:**
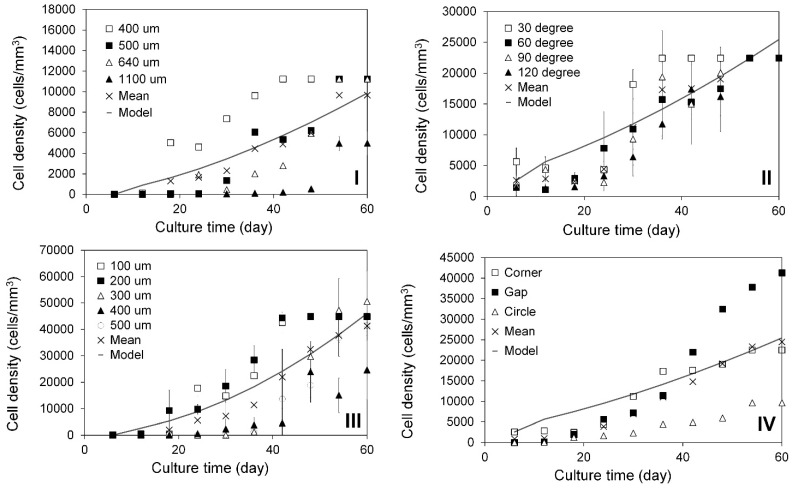
Rapid colonization of HDFs in finely controlled regular open structures following the slow colonization stage (6 days). HDF densities in (**I**) circular open pores (diameter: □: 400, ■: 500, △: 640, ▲: 1100 µm), (**II**) corners (angle: □: 30, ■: 60, △: 90, ▲: 120°), and (**III**) gaps (distance: □: 100, ■: 200, △: 300, ▲: 400, ○: 500 µm) on PLA discs, and (**IV**) the averaged cell densities in each (□: corners, ■: gaps, △: circular open pores) or all the open regular structures (×). All these rapid colonizations were simulated using power law models (solid line).

**Figure 5 ijms-25-02987-f005:**
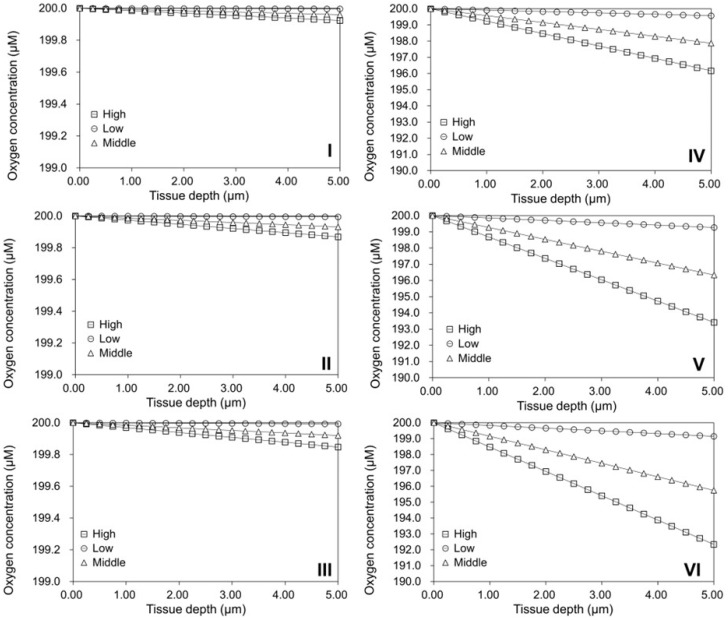
Simulation of oxygen concentrations within (**I**–**III**) HDFs or (**IV**–**VI**) hepatocytes cultured on spherical MSs (radius: 100 µm) for (**I**,**IV**) 0, (**II**,**V**) 7, or (**III**,**VI**) 14 days. For comparison purposes, high (□), low (○), and middle (△) values of oxygen consumption rate (OCR) of both cell types were used for the simulations, respectively.

**Figure 6 ijms-25-02987-f006:**
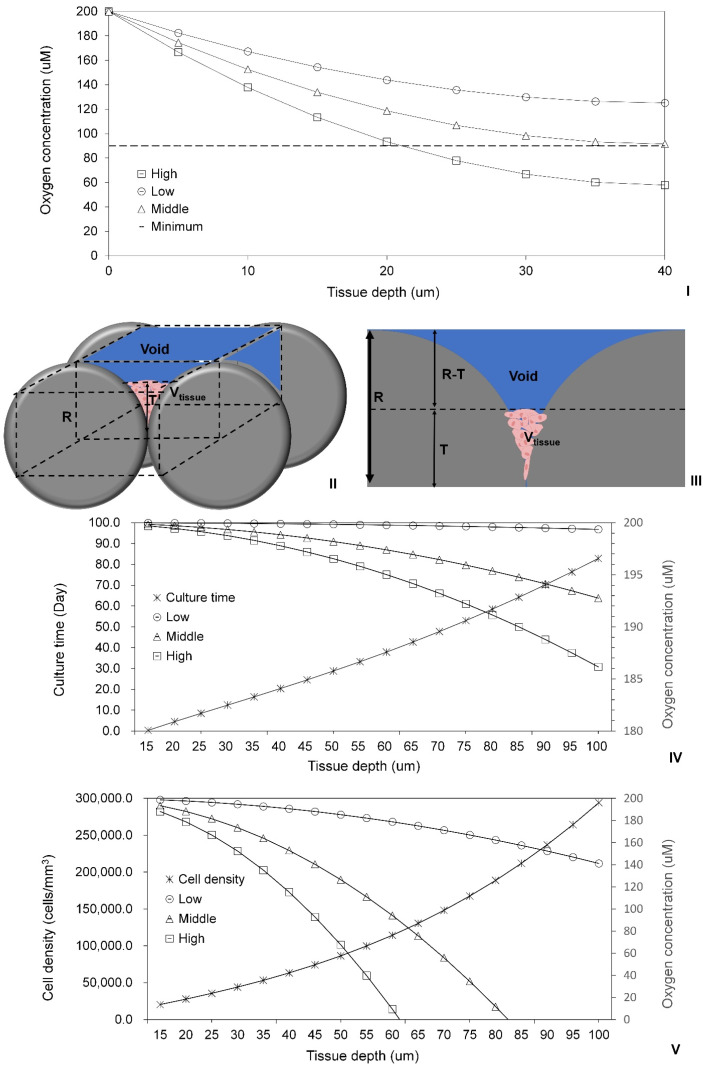
(**I**) Validation of the oxygen diffusion model using layered tissues. (--) The minimal oxygen concentration at the bottom of the layered human-endometrial-derived mesenchymal cells (hEMCs) to the maximum thickness of ~40 μm in Petri dishes was measured, and compared with the simulated oxygen concentrations from the top surface to the bottom of the 40 μm thick ECM sheet using high (□), low (○), and middle (△) values of OCR of hEMCs, respectively. (**II**) Schematic diagram and (**III**) cross-section of cells colonized within the void of the volume element selected in the tissue assembled using spherical MSs with the radius ® of 100 μm, where x is the depth into the tissue ranging from x = 0 at the surface and x = T at the bottom of the tissue; V_tissue_ is the volume of the colonized tissue. (**IV**,**V**) Simulation of oxygen concentrations within (**IV**) the human dermal fibroblasts (HDFs) or (**V**) hepatocytes colonized in the voids formed by two aggregated spherical MSs (radius: 100 µm). For comparison purposes, high (□), low (○), and middle (△) OCR values of both cell types were used for the simulations, respectively. The culture time of HDF (Ж) was predicted using the power law model in (**IV**), while the hepatocyte density (Ж) was calculated based on the reported hepatocyte volume in (**V**).

**Table 1 ijms-25-02987-t001:** Definitions and values of the constants used in the oxygen diffusion models.

Constant	Value	Reference
D ^1^	2.5 × 10^−9^ m^2^/s	[37]
C_0_ ^2^	2.0 × 10^−4^ M	[37]
C_min_(hMSCs) ^3^	9.0 × 10^−5^ M	[64]
OCR (hMSCs) ^4^	2.0 × 10^−17^ to 3.8 × 10^−17^ moles cell^−1^ s^−1^	[37]
OCR (human fibroblasts)	8.3 × 10^−19^ to 1.8 × 10^−17^ moles cell^−1^ s^−1^	[37]
OCR (human hepatocyte)	1.0 × 10^−16^ to 9.0 × 10^−16^ moles cell^−1^ s^−1^	[37]
R ^5^	100 × 10^−6^ m	[65]
Thickness of a fibroblast	4.7 μm	[62]
Volume of senescent human diploid fibroblast	2600 μm^3^	[66]
Volume of a hepatocyte	3400 μm^3^	[67]

^1^ D: diffusion coefficient of oxygen in a tissue or a hydrogel; ^2^ C_0_: concentration of oxygen in liquid media or a tissue; ^3^ C_min_: the minimum concentration of oxygen needed for human mesenchymal stem cells (hMSCs) to survive; ^4^ OCR: oxygen consumption rate; ^5^ R: radius of a microcarrier.

## Data Availability

Publicly available datasets were analyzed in this study.

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
