# Peer review of "Integrated Experimental and Mathematical Exploration of Modular Tissue Cultures for Developmental Engineering"

_ijms, 2024, doi:10.3390/ijms25052987_

Round 1

Reviewer 1 Report

Comments and Suggestions for Authors

Tissue engineering (TE) is a transformative field dedicated to the creation of biomimetic and functional tissues for medical applications.

In this study, the authors have tried to use a bottom-up approach to overcome the limitations of the top-down method in MT cultivation and assembly. In general, this study benefits from very good statistical, mathematical and experimental work and definitely deserves to be published in this high-level journal. However, there are points that should be corrected or clarified.

Comments about the content

1-      In the introduction part, the limitations and the achievement of this goal that it is necessary to use a new approach for MT cultivation and assembly, as well as the need to include diffusion equations in the modeling are well described and show the organized minds of authors.

2-       Although the introduction has a good text, it is surprising that, like the rest of the scientific articles, it lacks a paragraph that describes the literature review well. Definitely, adding these contents adds to the quality and completeness of the introduction section.

3-      Use the following references to deepen the introduction. 4D printing of PLA-TPU blends: effect of PLA concentration, loading mode, and programming temperature on the shape memory effect. Toughening PVC with Biocompatible PCL Softeners for Supreme Mechanical Properties, Morphology, Shape Memory Effects, and FFF Printability.

4-      There is an ambiguity in Figure 1 in the Section 2.1. It is said that one equation is considered for the upper section of the threshold and one for the lower section of the threshold. But the lines above the cell density threshold are broken. If an equation is considered for the upper section of the threshold, why this fracture exists near the value of 25000 cells/cm2. Explain this ambiguity.

5-      When a mathematical model is verified with experimental results, it is definitely necessary, except that the reader can see in the graph that these two trends are close to each other, there should also be goodness of fit values such as Mean absolute percentage error in the text. But unfortunately, these values are not visible.

6-      In the model development section, it is described that the equations related to diffusion are described by McMurtrey (2016). Assumptions are taken into account in solving equations, which play a major role in solving equations. But there is no convincing reason for this assumption. It is necessary to briefly explain the reason for each assumption so that the reader does not have to refer to the previous articles.

7-      Too wide and many equations are used in the article. It has been repeated many times in the text that the simulations have been done with a combination of different equations. It is suggested that all the final equations governing the problem be brought together in a small section and briefly explained. Definitely, adding this section can help the reader to understand the equations widely.

8-      Figures 2 and 6 are very large. When the figure is too large, the reader is forced to refer to the explanation below the figure frequently and back to the figure to understand the figures. It is better to divide each of these figures into two or more parts. Although the opinion of the authors in this field is definitely a priority.

Comments on the quality of English language

 It should be mentioned that the text has a good grammatical level

Author Response

General comments: Tissue engineering (TE) is a transformative field dedicated to the creation of biomimetic and functional tissues for medical applications. In this study, the authors have tried to use a bottom-up approach to overcome the limitations of the top-down method in MT cultivation and assembly. In general, this study benefits from very good statistical, mathematical and experimental work and definitely deserves to be published in this high-level journal. However, there are points that should be corrected or clarified.

Responses to the general comments: We highly appreciate the reviewer’s positive comments on the novelty of our study, particularly in regard to the integration of statistical, mathematical, and experimental methodologies for the analysis of modular tissue (MT) cultivation in the bottom-up DE approach. In response to the reviewer's valuable feedback, major revisions have been implemented to enhance the manuscript. We have provided detailed explanations for each point raised, addressing them individually to ensure clarity and scientific rigor.

Specific comment No.1: In the introduction part, the limitations and the achievement of this goal that it is necessary to use a new approach for MT cultivation and assembly, as well as the need to include diffusion equations in the modeling are well described and show the organized minds of authors.

Response to Point 1: We appreciate the reviewer's recognition of the strengths highlighted in our introduction section, particularly the thorough discussion of the limitations inherent in current methodologies and the necessity for a novel approach to MT cultivation and subsequent assembly. Additionally, the acknowledgment of the importance of incorporating diffusion equations into our modelling framework reflects our commitment to a comprehensive and organized approach.

Specific comment No.2: Although the introduction has a good text, it is surprising that, like the rest of the scientific articles, it lacks a paragraph that describes the literature review well. Definitely, adding these contents adds to the quality and completeness of the introduction section.

Response to Point 2: As per the constructive suggestion, a section of content has been incorporated between Lines 48-58, elucidating the concept of mass transfer limitations and emphasizing the significance of vascularization in tissue cultures. Detailed quantitative values have also been provided, including constraints on tissue size, diffusion distance, and oxygen consumption rate in both 2D cell and 3D tissue cultures, juxtaposed with those observed in natural mammalian tissues or the human body.

Specific comment No.3: Use the following references to deepen the introduction. 4D printing of PLA-TPU blends: effect of PLA concentration, loading mode, and programming temperature on the shape memory effect. Toughening PVC with Biocompatible PCL Softeners for Supreme Mechanical Properties, Morphology, Shape Memory Effects, and FFF Printability.

Response to Point 3: Upon careful consideration of the reviewer's recommendation, we have thoroughly reviewed these papers. Both works are commendable contributions to the research field of 3D-printing, with one focusing on 4D printing of PLA-TPU blends, specifically exploring the impact of PLA concentration, loading mode, and programming temperature on the shape memory effect; while the other investigates the toughening of PVC with biocompatible PCL softeners to enhance mechanical properties, morphology, shape memory effects, and FFF printability. While these papers are undoubtedly relevant to our study, their primary focus lies in material synthesis and 3D printing, areas that are of particular interest to the research group led by Dr. Andrew Gleadall at Loughborough University for enhancing their methodologies in 3D printing. However, it is important to note that our research utilized PLA discs kindly fabricated by Dr. Gleadall's group, with our primary focus being on cell and tissue cultures on these discs rather than the 3D printing process itself. Therefore, we have forwarded these papers to Dr. Gleadall's research team for their consideration, but have opted not to cite them in this manuscript.

Specific comment No.4: There is an ambiguity in Figure 1 in the Section 2.1. It is said that one equation is considered for the upper section of the threshold and one for the lower section of the threshold. But the lines above the cell density threshold are broken. If an equation is considered for the upper section of the threshold, why this fracture exists near the value of 25000 cells/cm2. Explain this ambiguity.

Response to Point 4: This is an insightful observation. Apparently, the reviewer meticulously examined both the 2D cell culture experiments and simulation outcomes. Our 2D cell culture experiments revealed that upon surpassing the threshold cell density (within the range of 6000-7000 cells/cm²), the cells adapted rapid growths. Specifically, as shown in Figure 1, when the cell density reached approximately 25000 cells/cm², exponential growth rates were observed, posing challenges in accurately quantifying cell populations due to the rapid density changes. This difficulty was compounded by the daily cell counting routines and limited monitoring time for each culture, leading to occasional discrepancies in growth curves, which was confirmed by our simulation results. Fortunately, our 2D cell culture experiments involved testing various initial cell seeding densities, and we found that employing a seeding density of 2700 cells/cm² enabled precise monitoring of cell density via our daily cell counting routine at day 8, yielding a count of approximately 25000 cells/cm².

Specific comment No.5: When a mathematical model is verified with experimental results, it is definitely necessary, except that the reader can see in the graph that these two trends are close to each other, there should also be goodness of fit values such as Mean absolute percentage error in the text. But unfortunately, these values are not visible.

Response to Point 5: Upon initiating the verification of experimental results using mathematical models, we encountered a dilemma. In the 2D cell culture experiments, we are limited to only two variables: cell density versus culture time. Consequently, employing complex models with multiple variables to simulate these 2D cell culture experiments and validate the models becomes exceedingly challenging. Thus, given the constraints of the experimental data consisting of only two limited variables, we resorted to utilizing relatively simple power law models to simulate these 2D cell culture results. However, due to the limited variables in the power law model (i.e. density and time), it could only describe the trend of cell growth rather than accurately simulate it. As noted by the reviewer, while the experimental and simulation graphs closely align to demonstrate similar trends, the simulation served merely as an approximation of the growth trend and was not precise enough for accurate simulation. Therefore, measures such as goodness of fit values, like Mean Absolute Percentage Error, were not calculated. As demonstrated in this manuscript, despite its simplicity, the power law models have enabled us to address various previously unconsidered issues and has enhanced our understanding of cell colonization in both 2D cell and 3D tissue cultures. Apart from the simulations, all the experimental results were analysed using one-way analysis of variance (ANOVA) for statistical significance. In order to develop the slow and rapid cell growth power law models, the 2D cell culture experimental data were particularly analysed using F-test and a two-tailed independent t-test as detailed in lines 202-210. A separate section of Statistical Analysis has been included in the revised version of the manuscript (lines 710-718).

 Specific comment No.6: In the model development section, it is described that the equations related to diffusion are described by McMurtrey (2016). Assumptions are taken into account in solving equations, which play a major role in solving equations. But there is no convincing reason for this assumption. It is necessary to briefly explain the reason for each assumption so that the reader does not have to refer to the previous articles.

Response to Point 6: As highlighted by the reviewer, in the section concerning oxygen diffusion model development, the equations pertaining to diffusion are elucidated based on the work of McMurtrey (2016). The assumptions underlying these equations significantly influence their solutions. To facilitate comprehension for the reader, these assumptions have been succinctly expounded upon within this manuscript (Line 644-652 and Line 669-674), obviating the need for reference to previous articles.

 Specific comment No.7: Too wide and many equations are used in the article. It has been repeated many times in the text that the simulations have been done with a combination of different equations. It is suggested that all the final equations governing the problem be brought together in a small section and briefly explained. Definitely, adding this section can help the reader to understand the equations widely.

 Response to Point 7: We appreciate the reviewer's observations regarding the extensive use of equations throughout this research. It has been duly noted that the simulations entail a combination of various equations, as reiterated multiple times within the text. Recognizing that the readership of this paper may encompass biologists with limited mathematical or simulation expertise, as well as modelers with limited cell culture experience, it was imperative to approach the model development in a gradual manner. Therefore, in the Materials and Methods section, we meticulously present the basic deviation process and assumptions step by step, aiming to ensure clarity and comprehension for readers with varying backgrounds. Additionally, all the final equations governing the problem are succinctly explained at the conclusion of their derivations. Furthermore, in response to the reviewer's suggestion, we have included a summary of the final equations governing the cell growth and oxygen

diffusion problems, accompanied by brief explanations of the variables to aid in reading and understanding between Lines 687-709.

Specific comment No.8:  Figures 2 and 6 are very large. When the figure is too large, the reader is forced to refer to the explanation below the figure frequently and back to the figure to understand the figures. It is better to divide each of these figures into two or more parts. Although the opinion of the authors in this field is definitely a priority.

Response to Point 8: We sincerely appreciate the reviewer's feedback. It has been noted that Figure 2 and Figure 6 appear larger compared to other figures in this manuscript. This is primarily attributed to the inclusion of schematic diagrams aimed at further elucidating the cultured cells within different culture environments. Considering that some readers may have limited or no experience in cell or tissue cultures, these diagrams are deemed crucial for facilitating their understanding of the significance of both experimental and simulated results. However, authors have strived to maintain clarity in the manuscript, as the inclusion of additional figures may potentially detract from its overall clarity and coherence. Furthermore, it has been observed that figures with similar size and format have been published in this journal previously. Therefore, it has been decided to keep the current format of these figures unless the reviewer insists on their separate presentation as crucial. 

Specific comment No. 9:  It should be mentioned that the text has a good grammatical level.

Response to Point 9: We sincerely value the reviewer's favorable remarks regarding the grammar and language utilized in this paper.

Reviewer 2 Report

Comments and Suggestions for Authors

This article is devoted to the study of an important problem associated with the development of effective tissue-engineered structures and is of interest to a wide range of specialists. The results obtained during the study are of scientific interest, but they cannot be recommended for practical use in the implementation of certain tissue engineering structures. Much additional research remains to be done to obtain more meaningful results. Nevertheless, after significant revision, the article may be recommended for publication. In this case, it is advisable to do the following.

1. It is not clear why the Material and Methods section is located after the Results and Discussions section. Typically, the first of above sections is placed before the second, which allows authors to reasonably explain to the reader the strategy for planning the theoretical and experimental parts of the study. But the main thing is that authors themselves have the opportunity to assess the prospect of the research and the significance of obtained results . In this work, despite the rather interesting results, they cannot be interpreted unambiguously, because they were obtained only taking into account subjectively selected parameters. For example, the quote: “To establish power law cell growth models through regression analyses, the 2D cell culture data were divided into slow and rapid growth phases using the threshold density (6500 cells/cm2). Density data within the range of 6500 ± 500 cells/cm2 were considered the threshold density and included in both slow and rapid growth phases" (lines 167-170). Question: why exactly 6500 cells/cm2? Why not 6000 cells/cm2 or 7000 cells/cm2? How will law cell growth change if a different threshold density is used in the model? Another quote: “In this study, experimental data with an initial cell seeding density of 265 cells/cm2 were employed to formulate the slow cell growth model... and the rapid cell growth model...” (lines 187-189). Question: will cell growth models change if the initial cell seeding density is changed? Authors, for example, state: “The simulation results indicate that the culture time required to produce MTs with specific target cell densities can be regulated by adjusting initial cell seeding densities” (lines 214-215).

There are quite a lot of similar subjective approaches that require clarification in the manuscript and all of them should be explained in detail.

2. Conclusions should be presented in more detail. Authors mainly focus on a well-known fact that they were able to confirm, namely: “The results highlighted the importance of manufacturing MTs with suitable cell densities, considering factors such as cell type, metabolic rate, initial seeding density, size and structure of individual and aggregated MSs, nutrient diffusion, and culture time period" (lines 660-663). At the same time, they note that “... our integrated experimental and computational methods could potentially inform bioprocess design for large-scale MT manufacture and propose suitable strategies for assembling MTs into more functional 3D tissues in DE” (lines 663- 666). That is, in fact, they state the fact that in order to obtain reliable, practically significant results, it is to carry out a fairly large amount of research yet.

3. In lines 601 and 605 McMurtrey (2016) replace with [33].

4. In line 604 “Where” should be written with a small letter – “where”.

Author Response

General Comments and Suggestions for Authors: This article is devoted to the study of an important problem associated with the development of effective tissue-engineered structures and is of interest to a wide range of specialists. The results obtained during the study are of scientific interest, but they cannot be recommended for practical use in the implementation of certain tissue engineering structures. Much additional research remains to be done to obtain more meaningful results. Nevertheless, after significant revision, the article may be recommended for publication. In this case, it is advisable to do the following.

Response to the general comments: We express our sincere gratitude for the reviewer's constructive feedback on our research. This article addresses a significant problem concerning the development of effective tissue-engineered structures and is anticipated to be of interest to a diverse audience of specialists. While the culture systems (such as the PLA discs and the PMMA spherical particles), as well as the simulation methods (such as the simple power law models) employed in this study primarily pertain to issues in tissue engineering and developmental engineering, and offer mechanistic insights into 3D tissue cultures, modular tissue cultures and subsequent tissue assembly, it is noted that these systems and models may not directly translate to practical applications in tissue engineering, as suggested by the reviewer. However, the research findings, particularly regarding cell seeding efficiency in modular tissue cultures and its impact on culture times and cell populations within the cultured tissues, hold great practical value for developmental engineering applications. We concur with the reviewer's assessment that further research is necessary to enhance our understanding of cell colonization in modular scaffolds and the subsequently assembled tissues. Accordingly, major revisions have been undertaken based on the thoughtful comments provided by the reviewers, and we trust that the current version of the manuscript is suitable for publication. Below, we provide detailed responses to specific comments raised by the reviewer.

Specific comment No.1:   It is not clear why the Material and Methods section is located after the Results and Discussions section. Typically, the first of above sections is placed before the second, which allows authors to reasonably explain to the reader the strategy for planning the theoretical and experimental parts of the study. But the main thing is that authors themselves have the opportunity to assess the prospect of the research and the significance of obtained results. In this work, despite the rather interesting results, they cannot be interpreted unambiguously, because they were obtained only

taking into account subjectively selected parameters. For example, the quote: “To establish power law cell growth models through regression analyses, the 2D cell culture data were divided into slow and rapid growth phases using the threshold density (6500 cells/cm2). Density data within the range of 6500 ± 500 cells/cm2 were considered the threshold density and included in both slow and rapid growth phases" (lines 167-170). Question: why exactly 6500 cells/cm2? Why not 6000 cells/cm2 or 7000 cells/cm2? How will law cell growth change if a different threshold density is used in the model? Another quote: “In this study, experimental data with an initial cell seeding density of 265 cells/cm2 were employed to formulate the slow cell growth model... and the rapid cell growth model...” (lines 187-189). Question: will cell growth models change if the initial cell seeding density is changed? Authors, for example, state: “The simulation results indicate that the culture time required to produce MTs with specific target cell densities can be regulated by adjusting initial cell seeding densities” (lines 214-215). There are quite a lot of similar subjective approaches that require clarification in the manuscript and all of them should be explained in detail.

Response to Point 1: We appreciate the reviewer's comments. The writing format of our manuscript adheres to the template recommended by the journal, which places the Materials and Methods section after the Results and Discussions. As elucidated in the manuscript, the results presented in this research stem from cell and tissue experiments (i.e. 2D cell cultures on flat tissue culture surfaces, and tissue cultures within the regular open structures of the 3D-printed PLA discs) that were subsequently simulated, with parameters (such as reaction or cell growth order (n) and reaction or cell growth constant (k) in the power law models) were calculated and selected objectively using the experimental data, rather than selected subjectively. For instance, the threshold density range of 6000-7000 cells/cm² was determined through our 2D cell culture experiments, acknowledging that it may not precisely be 6500 cells/cm², but rather could fall within the specified range of 6000 cells/cm² to 7000 cells/cm². However, for the sake of clarity and ease of simulation, we opted to utilize the midpoint value of the range, i.e., 6500 cells/cm², in this research. When the initial cell density fell below this threshold range, as depicted in Figure 1, the cells exhibited slow growths, thus necessitating the use of a slow growth model for simulation. Conversely, when the initial cell density surpassed the threshold range, as also illustrated in Figure 1, the cells exhibited accelerated growths, leading to the employment of quick growth model for simulation. Hence, the selection of the growth model was also grounded in the experimental findings. The power law models were initially developed based on 2D cell culture data and subsequently extrapolated to predict cell growth on the solid surfaces of 3D scaffolds. Similarly, the power law models were also developed based on cell colonization within the regular open structures in the 3D-printed PLA discs, and subsequently extrapolated to predict cell colonization within the irregular open structures in normal 3D porous scaffolds such as cellulosic scaffolds. As aptly noted by the reviewer, "The simulation results indicate that the culture time required to produce MTs with specific target cell densities can be regulated by adjusting initial cell seeding densities" (Lines 237-239). These results were derived from model simulations, calculation and analysis using the power law models, as explicated in various sections of the manuscript, and were not derived from subjective approaches. That is, during MT cultures, both initial cell seeding density and culture time can be adjusted to achieve certain cell density on the surfaces of the modular scaffolds, as calculated using the corresponding slow or rapid cell growth power law models.

Specific comment No.2:   Conclusions should be presented in more detail. Authors mainly focus on a well-known fact that they were able to confirm, namely: “The results highlighted the importance of manufacturing MTs with suitable cell densities, considering factors such as cell type, metabolic rate, initial seeding density, size and structure of individual and aggregated MSs, nutrient diffusion, and culture time period" (Lines 729-732 in the revised version). At the same time, they note that “... our integrated experimental and computational methods could potentially inform bioprocess design for large-scale MT manufacture and propose suitable strategies for assembling MTs into more functional 3D tissues in DE” (Lines 732-735 in the revised version). That is, in fact, they state the fact that in order to obtain reliable, practically significant results, it is to carry out a fairly large amount of research yet.

Response to point 2: We acknowledge the reviewer's comments regarding the need for more detailed conclusions. In the conclusion section, we initially underscored the elements/factors that our research confirms, such as "The results highlighted the importance of manufacturing MTs with suitable cell densities, considering factors such as cell type, metabolic rate, initial seeding density,

size and structure of individual and aggregated MSs, nutrient diffusion, and culture time period." (Lines 729-732) Subsequently, we emphasized that "...our integrated experimental and computational methods could potentially inform bioprocess design for large-scale MT manufacture and propose suitable strategies for assembling MTs into more functional 3D tissues in DE." (Lines 732-735). This assertion stems from the recognition that optimizing cell seeding methods to efficiently seed cells on the surfaces of modular scaffolds while minimizing culture time is crucial and cost-effective for producing large quantities of MTs at a very large scale. MTs with appropriate cell densities will facilitate subsequent assembly procedures to produce more functional 3D tissues. Therefore, this research clearly holds the potential to inform the bioprocess design for large-scale MT manufacture and propose suitable strategies for assembling multiple MTs into more functional 3D tissues in DE, which has been modified in the current version of the conclusion section.

Specific comment No.3: In lines 601 and 605 McMurtrey (2016) replace with [33].

Response to point 3: As per the kind suggestion, reference number [37] (updated after adding extra references) has been substituted for McMurtrey (2016), as indicated in Line 635-636 and Lines 640-641 of the manuscript.

Specific comment No.4: In line 604 “Where” should be written with a small letter – “where”.

Response to point 4: As kindly suggested, “Where” has been replaced by “where” for correct use in Line 639.

Reviewer 3 Report

Comments and Suggestions for Authors

The subject of this manuscript is using an integrated experimental and computational approach to (a) Develop power-law models to simulate cell growth on surfaces and porous scaffolds. (b) Adapt an oxygen diffusion model to predict concentrations in modular tissues. (c) Combine models to provide insights on manufacturing modular tissues and assembling them into larger tissues through a developmental engineering strategy.

1.       The explanation of oxygen diffusion limitations in conventional TE remains vague. Be more specific on maximal tissue thickness and hypoxia issues. Cite values from the literature. 

2.       More references are needed to substantiate claims about vascularisation issues and the complexity limitations of conventional TE.

3.       Explanation of the developmental engineering concept is still very cursory. It should have more details on the biomimetics of developmental processes.

4.       The abstract mentions that power law models were developed to simulate cell growth, and an oxygen diffusion model was adapted. However, the methodology section provides insufficient details on the model development and adaptation process. More information is needed on the model equations, assumptions, inputs, outputs, etc.

5.       Details on material preparation/fabrication are inadequate - need full descriptions of procedures and parameters.

6.       The statistical analysis section requires more details - specify the exact tests used for model development, validation, etc.

7.       Sandwiching a methods section between results and discussion is an unorthodox structure. It should follow the typical IMRAD format.

8.       In the results section, the presentation of the data and modelling outputs is very text-heavy. More figures/graphs should be added to effectively visualise experimental data, model fits, oxygen concentration profiles, etc. This will significantly improve readability.

9.       Data presentation and visualisation remain weak despite being the main results of research. Significant opportunity for graphs and figures.

10.   Linkages of experimental data to modelling outputs are not clearly described. This connection needs strengthening. 

11.   Discussion of context and implications can go much deeper. Limitations of the current study and models need highlighting.

12.   The claim that this methodology can inform bioprocess design for modular tissue manufacture and upscaling requires more substantiation. Can some case studies or examples be provided to demonstrate this?

13.   In conclusion, the long, wordy sentences hamper readability. Break these up.

14.   Claims around guiding bioprocess design are overreaching based on the evidence presented. Scale back to reflect the scope of the study.

15.   Broader impacts and future opportunities can be elaborated on more.

16.   Were any statistical tests conducted during data analysis and modelling? This should be clearly described.

17.   What sensitivity analyses were performed for the models? How robust are the models to changes in parameters or inputs?

18.   Have the models been validated with independent experimental data sets beyond what was used for initial development? 

19.   What plans exist to validate model predictions on modular tissue assembly and upscaling experimentally?

20.   Visual presentation of growth data and modelling outputs is inadequate. More graphical depictions would enhance communication and understanding tremendously.

Comments on the Quality of English Language

·         Many long, wordy sentences affect readability - these should be shortened for clarity. Examples include the run-on sentences in the Introduction and Conclusion sections.

·         There are several typos in technical terms, abbreviations, chemical formulas, etc, throughout the text. Meticulous proofreading is a must not just for grammar but also for factual accuracy.

·         Ensure British/American English spellings are consistent rather than used interchangeably. For example, "modeled" vs "modelled". Pick one format.

·         While the content appears sound, polishing the English writing style via proofreading will elevate this manuscript greatly. 

Author Response

General Comments and Suggestions for Authors: The subject of this manuscript is using an integrated experimental and computational approach to (a) Develop power-law models to simulate cell growth on surfaces and porous scaffolds. (b) Adapt an oxygen diffusion model to predict concentrations in modular tissues. (c) Combine models to provide insights on manufacturing modular tissues and assembling them into larger tissues through a developmental engineering strategy.

Responses to the general comments: We sincerely appreciate the reviewer's constructive feedback. The focus of this manuscript lies in employing an integrated experimental and computational approach to develop power-law models for simulating cell growth on surfaces and porous scaffolds. Subsequently, we adapt an oxygen diffusion model to predict concentrations in modular tissues. Finally, we combine these models to offer insights into manufacturing modular tissues (MT) and assembling them into larger tissues through a developmental engineering strategy. Below, we provide detailed responses to the specific points raised by the reviewer.

Specific comment No.1:  The explanation of oxygen diffusion limitations in conventional TE remains vague. Be more specific on maximal tissue thickness and hypoxia issues. Cite values from the literature. 

Response to Point 1: As kindly suggested, a concise description has been incorporated between Lines 48-58, highlighting factors such as limited tissue size, restricted diffusion distance, and oxygen consumption rate in 2D cell culture and 3D tissue culture when compared to natural mammalian tissues. Specific values have been provided to ensure clarity regarding mass transfer limitations and the significance of vascularization in tissue cultures.

Specific comment No.2:  More references are needed to substantiate claims about vascularisation issues and the complexity limitations of conventional TE.

Response to Point 2: As recommended, additional references have been included in Line 48-58, Line 59 and Line 64 to further substantiate the assertion regarding conventional tissue engineering limitations.

Specific comment No.3:  Explanation of the developmental engineering concept is still very cursory. It should have more details on the biomimetics of developmental processes.

Response to Point 3: As requested, further details have been provided in Line 84-90 to offer a more comprehensive explanation of the developmental engineering concept. Additionally, more information on the biomimetics of developmental processes has been included to enhance understanding.

Specific comment No.4:  The abstract mentions that power law models were developed to simulate cell growth, and an oxygen diffusion model was adapted. However, the methodology section provides insufficient details on the model development and adaptation process. More information is needed on the model equations, assumptions, inputs, outputs, etc.

Response to Point 4: As addressed in response to the 1st reviewer’s comments, we acknowledge the diverse readership of this paper, which includes biologists with limited mathematical or simulation expertise, as well as modelers with limited cell culture experience. Thus, it is essential to provide sufficient details of the model development procedure in a gradual manner. Furthermore, we also appreciate the observations of the 1st reviewer regarding the extensive use of equations throughout the article. To address this concern and as kindly suggested by this reviewer, we have meticulously presented the basic derivation process and assumptions step by step in the Materials and Methods section. This approach aims to ensure clarity and comprehension for readers with varying backgrounds. Moreover, all the final equations governing the problem are succinctly explained at the conclusion of their derivations. Additionally, we have included a summary of the final equations governing the problems, along with brief explanations, to facilitate reading and understanding in lines 687-709.

Specific comment No.5:  Details on material preparation/fabrication are inadequate - need full descriptions of procedures and parameters.

Response to Point 5: Given the utilization of both experimental and computational methods in this study, it is imperative to furnish sufficient details for all experimental and simulation procedures, enabling reproducibility and utilization by other colleagues. As previously mentioned, additional details have been incorporated into the model development sections. Consequently, details regarding material preparation/fabrication should be presented as concisely as possible. As recommended, a brief outline of the particle preparation and fabrication procedure has been supplemented between Lines 525-527, while 3D printing parameters have been included between Lines 540-543. Considering that the materials fabrication aspect is not the primary focus of this study, more relevant detailed information regarding material preparation/fabrication can be found in the provided reference.

Specific comment No.6:  The statistical analysis section requires more details - specify the exact tests used for model development, validation, etc.

Response to Point 6: As previously discussed in response to the 1st reviewer, our 2D cell culture experiments are inherently constrained by only two variables: cell density versus culture time. Consequently, employing complex models with multiple variables to simulate and validate these experiments poses significant challenges. Therefore, due to the limitations of the experimental data consisting of only these two variables, we opted to utilize power law models to simulate the culture results. However, given the limited variables in the power model (density and time), it could only capture the trend of cell growth rather than precisely simulate it. As noted by the 1st reviewer, while the experimental and simulation graphs closely align to illustrate similar trends, the simulation serves as an approximation of the growth trend and lacks the precision for accurate simulation. Hence, measures such as statistical analysis for model development and validation, as well as goodness of fit values, were not conducted. The primary purpose of the model was to further validate the adaptation of slow or quick cell growths based on the initial cell seeding densities. Despite its simplicity, the

model has enabled us to address various previously unexplored issues and has enhanced our understanding of cell colonization in both 2D and 3D cell cultures, as demonstrated in this manuscript. Apart from the simulations, all the experimental results were analysed using one-way analysis of variance (ANOVA) for statistical significance. In order to develop the slow and rapid cell growth power law models, the 2D cell culture experimental data were particularly analysed using F-test and a two-tailed independent t-test as detailed in Lines 202-210. A separate section of Statistical Analysis has been included in the revised version of the manuscript in Lines 710-718.

Specific comment No.7:  Sandwiching a methods section between results and discussion is an unorthodox structure. It should follow the typical IMRAD format.

Response to Point 7: We fully appreciate this comment. However, the suggested template format advises placing the Materials and Methods section after the Results and Discussions section.

Specific comment No.8:  In the results section, the presentation of the data and modelling outputs is very text-heavy. More figures/graphs should be added to effectively visualise experimental data, model fits, oxygen concentration profiles, etc. This will significantly improve readability.

Response to Point 8:  We fully agree with the reviewer’s comment, thus in the preparation of this manuscript, various figures and graphs have been used to effectively visualise and explain experimental data and the simulation results to improve the readability. It is recognized that this paper caters to a diverse readership, encompassing biologists with limited mathematical or simulation expertise, as well as modelers with limited cell culture experience. Hence, it is imperative to provide sufficient details and explanations of the experimental and simulation results. In addition to textual explanations, schematic diagrams have been included in Figures 1, 2, 3 and 6. These diagrams serve to elucidate different cell culture, tissue environments, the assembly of MTs with different densities and its influences of vascularization and cell survival. Moreover, these diagrams can effectively visualize experimental and simulation data, and illustrate the influences of factors such as cell density, culture time, and oxygen concentration profiles on MT cultures and subsequent assembly into large 3D functional tissues. Apart from the combination of schematic diagrams with figures, the simulation results and the experimental results are demonstrated and compared in Figure 1 and Figure 4, while various detailed simulations were provided in Figure 5 for the ease of comparisons. We believe that these visual aids significantly enhance readability and understanding of the findings in this study.

Specific comment No.9:  Data presentation and visualisation remain weak despite being the main results of research. Significant opportunity for graphs and figures.

Response to Point 9: We wholeheartedly concur with the reviewer's perspective that data presentation and visualization are integral components of research outcomes. There is a notable emphasis on the utilization of various forms of graphical representations and visual aids. In addition to textual explanations, our manuscript incorporates various microscopic images, plots, and schematic diagrams, notably in Figures 1, 2, 3 and 6. Apart from the combination of schematic diagrams with figures, the simulation results and the experimental results are demonstrated and compared in Figure 1 and Figure 4, while various detailed simulations were provided in Figure 5 for the ease of comparisons. We believe these diverse methods of presenting data and visualizations are pivotal in elucidating different cell culture environments, effectively depicting experimental data, and illustrating the impacts of factors such as cell density, culture time, etc., on MT cultures and subsequent assembly processes.

Specific comment No.10: Linkages of experimental data to modelling outputs are not clearly described. This connection needs strengthening. 

Response to Point 10: We recognize the importance of establishing links between experimental data and model development and validation. Consequently, 2D cell culture experiments were conducted to

ascertain the relationship between cell growth and initial cell seeding density, which served as the basis for developing the power law models. Similarly, the experimental data of cell colonization within the regular open structures of PLA discs, were used to develop power law models for 3D tissue cultures. These connections between experimental data and modeling development are elaborated and reinforced in different sections of the manuscript such as Lines 189-216 and Lines 314-325. Subsequently, the developed power models underwent confirmation, validation, and further discussion through cell cultures conducted on modular scaffolds. The associations between experimental data and modeling output are also elucidated and strengthened in various sections such as Lines 371-373 and Lines 461-473.

Specific comment No.11:   Discussion of context and implications can go much deeper. Limitations of the current study and models need highlighting.

Response to Point 11: As recommended, the discussion regarding context and implications has been further revised. Specifically, the limitations of the current study and models are emphasized in Line 499-506.

Specific comment No.12:  The claim that this methodology can inform bioprocess design for modular tissue manufacture and upscaling requires more substantiation. Can some case studies or examples be provided to demonstrate this?

Response to Point 12: As addressed in response to the 2nd reviewer’s comments, in the conclusion section, we underscored that "...our integrated experimental and computational methods have the potential to inform bioprocess design for large-scale MT manufacture and suggest appropriate strategies for assembling MTs into more functional 3D tissues in DE." (Line 732-735). This assertion is grounded in the recognition that optimizing cell seeding methods to effectively seed cells on the surfaces of modular scaffolds while minimizing culture time is essential and cost-effective for producing significant quantities of MTs on a large scale. MTs with suitable cell densities will facilitate subsequent assembly procedures to generate more functional 3D tissues. Thus, this research distinctly offers the potential to inform and shape bioprocess design for large-scale MT manufacture and propose suitable strategies for assembling MTs into more functional 3D tissues in DE.

Specific comment No.13:  In conclusion, the long, wordy sentences hamper readability. Break these up.

Response to Point 13: As recommended, certain sentences in the conclusion, introduction and other sections have been divided into shorter ones, as seen in Lines 729-735, Lines 31-44, Lines 61-74, and Lines 102-107.

Specific comment No.14:  Claims around guiding bioprocess design are overreaching based on the evidence presented. Scale back to reflect the scope of the study.

Response to Point 14: As addressed in point 12 of our response, this assertion is based on the acknowledgment that optimizing cell seeding methods to efficiently seed cells on modular scaffold surfaces while minimizing culture time is crucial and cost-effective for producing substantial quantities of MTs on a large scale. MTs with appropriate cell densities will enable more effective assembly procedures, ultimately leading to the creation of more functional 3D tissues. Therefore, this research holds the potential to inform or influence bioprocess design for large-scale MT manufacture and propose effective strategies for assembling MTs into more functional 3D tissues in DE.

Specific comment No.15:  Broader impacts and future opportunities can be elaborated on more.

Response to Point 15: We concur with the reviewers' comments, and therefore, in the conclusion, we emphasize that our integrated experimental and computational methods hold the potential to inform or guide bioprocess design for large-scale MT manufacture and propose suitable strategies for

assembling MTs into more functional 3D tissues in DE, as addressed in specific points 12 and 14 of our responses.

Specific comment No.16.   Were any statistical tests conducted during data analysis and modelling? This should be clearly described.

Response to Point 16: As we responded to specific point 6, all the experimental results were analysed using one-way analysis of variance (ANOVA) for statistical significance. In order to develop the slow and rapid cell growth power law models, the 2D cell culture experimental data were particularly analysed using F-test and a two-tailed independent t-test as detailed in lines 202-210. A separate section of Statistical Analysis has been included in the revised version of the manuscript (lines 710-718).

Specific comment No.17.   What sensitivity analyses were performed for the models? How robust are the models to changes in parameters or inputs?

Response to Point 17: In this study, various cell seeding densities were employed to assess the dependency of cell growth on initial cell seeding densities and to conduct sensitivity analyses regarding slow and rapid cell growths in 2D cell cultures. As explained in response to point 6, due to the limited variables in the power model (density and time), the power law cell growth models could only capture the trend of cell growth rather than precisely simulate it. Consequently, measures such as statistical analysis for model development and validation, as well as goodness of fit values, were not conducted. Nonetheless, the power law cell growth models were utilized to simulate cell growths based on different cell seeding densities, which were confirmed by experimental data and also used for sensitivity analyses. Moreover, the oxygen diffusion model was used to simulate and compare HDFs and Hepatocytes with various OCR values for comparison purposes and also sensitivity analyses.

Specific comment No.18:  Have the models been validated with independent experimental data sets beyond what was used for initial development? 

Response to Point 18: The 2D culture of HDFs using various tissue culture plates and flasks is a common practice for postgraduate and undergraduate research projects in our group. In these independent cell culture experiments, cell densities are regularly monitored at different time periods, which have been utilized to further validate the developed slow and rapid cell growth power law models even after the submission of this manuscript.

Specific comment No.19:    What plans exist to validate model predictions on modular tissue assembly and upscaling experimentally?

Response to Point 19: Compared to 2D cell cultures on the flat surfaces of tissue culture plates or flasks, accurately counting cells on the curved or rough surfaces of modular scaffolds and 3D porous scaffolds, and the cells colonized inside the regular and irregular open pores of the PLA discs and other 3D porous scaffolds using conventional phase contrast or fluorescent microscopes poses challenges. Therefore, our intention is to develop new methods for accurately quantifying cells on these surfaces and open pores through DAPI staining and assay. This approach can be utilized to validate model predictions in large-scale modular tissue cultures and further tissue assembly processes.

Specific comment No.20:  Visual presentation of growth data and modelling outputs is inadequate. More graphical depictions would enhance communication and understanding tremendously.

Response to Point 20: As addressed in point 9 of our response, it is crucial to utilize a variety of graphical representations and visual aids. Alongside textual explanations, our manuscript includes

various microscopic images, plots, and schematic diagrams, particularly in Figures 1, 2, 3 and 6. These diverse approaches to presenting data and visualizations are essential in elucidating various cell culture environments, effectively illustrating experimental data, and demonstrating the effects of factors such as cell density and culture time on MT cultures and subsequent assembly processes.

Specific comment No.21:  Comments on the Quality of English Language. · Many long, wordy sentences affect readability - these should be shortened for clarity. Examples include the run-on sentences in the Introduction and Conclusion sections. · There are several typos in technical terms, abbreviations, chemical formulas, etc, throughout the text. Meticulous proofreading is a must not just for grammar but also for factual accuracy. · Ensure British/American English spellings are consistent rather than used interchangeably. For example, "modeled" vs "modelled". Pick one format. · While the content appears sound, polishing the English writing style via proofreading will elevate this manuscript greatly. 

Responses to Point 21: Despite the first reviewer's positive feedback on the grammatical level of this manuscript, we have taken the kind comments into consideration and made some modifications. We have conducted a major revision of the manuscript, particularly cut the long sentences into even shorter ones in various sections such as Lines 31-44, lines 61-74, lines 102-107, Lines 729-735, to improve readability and understanding. Additionally, as suggested, we have adopted consistent American English spelling throughout the manuscript. For example, we have replaced all instances of "modelled" with "modeled" to maintain consistency.

Round 2

Reviewer 1 Report

Comments and Suggestions for Authors

Accept.

Reviewer 2 Report

Comments and Suggestions for Authors

I agree with corrections made by authors in the manuscript after the first stage of review.